## PROCEEDINGS A

computer modelling and simulation, geophysics, statistics

hazard assessment, sediment amplification, unstructured mesh, Karachi port, Makran subduction zone, coastal engineering

**Author for correspondence:**
Devaraj Gopinathan
e-mail: d.gopinathan@ucl.ac.uk

# Probabilistic quantification of tsunami current hazard using statistical emulation

Devaraj Gopinathan[1], Mohammad Heidarzadeh[2] and Serge Guillas[1]

[1]Department of Statistical Science, University College London, Gower Street, London WC1E 6BT, UK
[2]Department of Civil and Environmental Engineering, Brunel University London, Uxbridge UB8 3PH, UK

DG, 0000-0002-0490-3229

In this paper, statistical emulation is shown to be an essential tool for the end-to-end physical and numerical modelling of local tsunami impact, i.e. from the earthquake source to tsunami velocities and heights. In order to surmount the prohibitive computational cost of running a large number of simulations, the emulator, constructed using 300 training simulations from a validated tsunami code, yields 1 million predictions. This constitutes a record for any realistic tsunami code to date, and is a leap in tsunami science since high risk but low probability hazard thresholds can be quantified. For illustrating the efficacy of emulation, we map probabilistic representations of maximum tsunami velocities and heights at around 200 locations about Karachi port. The 1 million predictions comprehensively sweep through a range of possible future tsunamis originating from the Makran Subduction Zone (MSZ). We rigorously model each step in the tsunami life cycle: first use of the three-dimensional subduction geometry Slab2 in MSZ, most refined fault segmentation in MSZ, first sediment enhancements of seabed deformation (up to 60% locally) and bespoke unstructured meshing algorithm. Owing to the synthesis of emulation and meticulous numerical modelling, we also discover substantial local variations of currents and heights.

# 1. Introduction

Following the unexpected damage incurred at ports from the tsunamis of 2004 (Indian Ocean), 2010 (Chile) and 2011 (Japan) [1,2], it is paramount to investigate the associated hazard. Despite recent studies [1,3–5] and advances in high-fidelity modelling [6], probabilistic methods for quantification of future tsunami hazard due to strong flows in harbours are sparse [7,8]. The need for such a quantification is further accentuated by certain peculiarities related with the phenomena of large tsunami currents in ports, e.g. the drifting of the 285 m ship *Maersk Mandraki* on 26 December 2004 at the Omani port of Salalah [2], despite small wave heights. It is deceptive to associate high wave amplitudes with high velocities. The treacherous nature of the currents was aggravated by the fact that strong currents in harbours continued for hours after the waves with maximum amplitude had arrived. This is all the more consequential since conventional tsunami warnings may be lifted after visibly perceptible signs of the tsunami (i.e. vertical displacement) have disappeared, whereas the strong currents may manifest later on.

Probabilistic scenario-based tsunami hazard assessment (PTHA) delivers *a priori* critical data to buttress tsunami disaster planning and practice. Scenario-based assessment scores over its catalogue-based counterpart through a more comprehensive exploration of plausible scenarios. Probabilistic scenario-based assessment surpasses deterministic scenario-based assessment in its assignment of probabilities and weighed integration of the different plausible scenarios. There exist variants in the probabilistic methodologies employed in PTHA: Monte Carlo [9], logic-tree [10] and Bayesian [11]. For an in-depth discussion on PTHA, the reader is referred to the recent review of Grezio *et al.* [12]. However, apart from the difficulties in assigning probabilities to scenarios, the computational burden expended for simulating each scenario prohibits an exhaustive sweep over the entire range of plausible scenarios. In this work, we pursue another probabilistic route via statistical emulation to quantify uncertainties in future tsunamis due to the uncertain earthquake sources (see workflow in figure 1*a*). Since the probability of large magnitude events is small, a comprehensive coverage of the Gutenberg–Richter (G–R) relation requires a large number of runs for the diversity of plausible events to be well represented across magnitudes and source locations (at least thousands for a coarse quantification and orders more for realistic assessments). Due to the considerable computational complexity of each simulation of coastal tsunami currents, such a probabilistic endeavour is made feasible by essentially replacing the numerical tsunami model by a statistical surrogate: the emulator. To our knowledge, this is the first time that Gaussian process (GP) emulation has been marshalled to generate future earthquake-generated tsunami currents; it has been employed only once in the past for currents, for a single source of landslide-generated tsunamis with considerable benefits in terms of computational costs and hazard assessment [13]. Here, with a design of only 300 full-fledged training runs, we fit an emulator to rapidly predict the impact of 1 million plausible tsunamis at prescribed locations. These emulated runs enable us to characterize uncertainty in future tsunami currents. A recent work by Kotani *et al.* [14] adopts a similar strategy of approximating the input–output response surface, albeit using nonlinear regression. Zhang & Niu [15] showcase a comparable 1.38 million scenarios, although using linear combination of waves from unit sources. Another recent strategy for reduction of the number of tsunami simulations employs an event tree coupled with cluster filtering of sources [16,17].

Additionally, formidable computational challenges must be addressed in order to accurately represent both the actual geophysical processes and their uncertainties. Despite possible issues arising from handling fine resolutions, our main challenge lies in encapsulating a large number of these high-definition simulations within a statistical framework. This is an essential requirement for PTHA and stretches the limit of current high-performance computing (HPC) facilities, even with the latest graphics processing unit (GPU) acceleration [18]. Often, the trade-off between capability and capacity in HPC is left unresolved by either radically simplifying the physics (e.g. a linear tsunami propagation till say 100 m depth with the use of an empirical relationship thereafter), or running very few fine resolution simulations as scenarios. Given a validated tsunami model, we argue that our emulation framework, in this context of currents that are

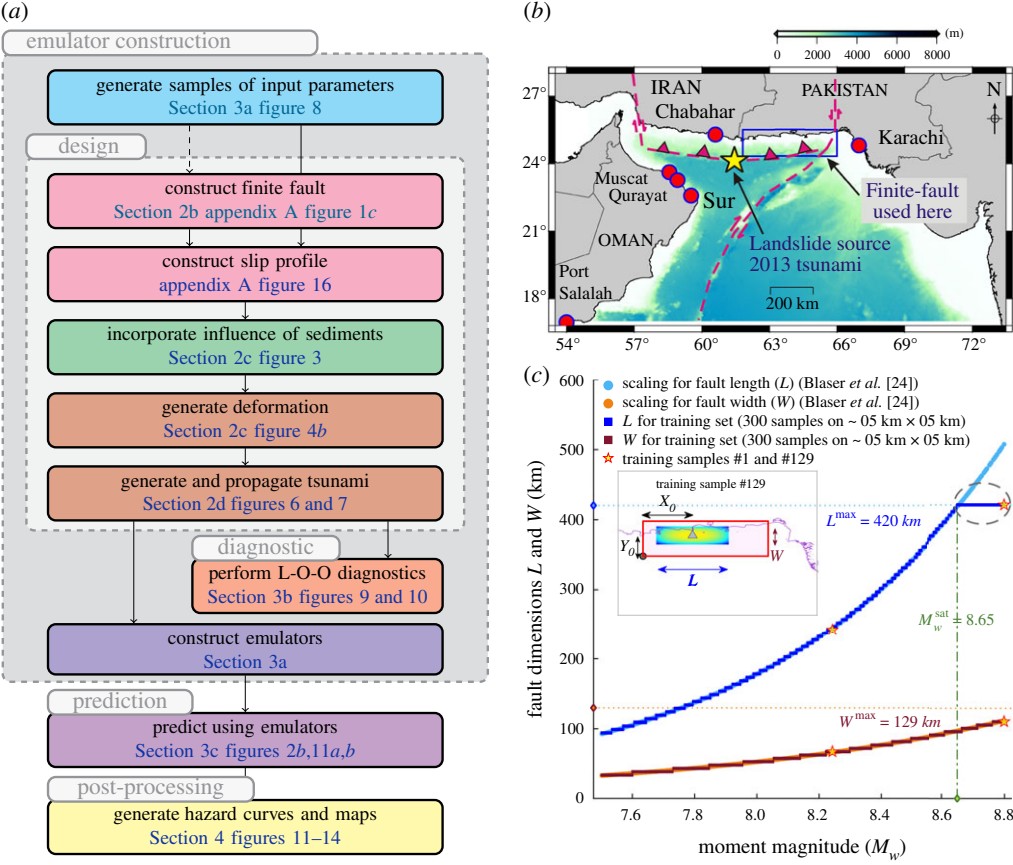

**Figure 1.** (*a*) Global workflow describing the integration of different work components in this study (see electronic supplementary material for a more detailed workflow). (*b*) The Makran Subduction Zone (MSZ). (*c*) Fault dimensions, i.e. length (*L*) and width (*W*) of 300 earthquake scenarios plotted over the scaling relation with respect to the moment magnitude ($M_w$). It shows the maximum length ($L_{max}$), width ($W_{max}$) and moment magnitude ($M_w^{sat}$) accommodable in the eastern MSZ. *L* saturates (ellipse) after $M_w$ 8.65 (green line). The inset shows the fault dimensions (*L*, *W*) and epicentre coordinates ($X_0$, $Y_0$) for scenario no. 129. (Online version in colour.)

nonlinear and very sensitive to near shore bathymetry, attempts a solution to this trade-off between precision and coverage of uncertainties. It requires manipulation of very large datasets on HPC, as well as complex post-processing on diverse software and data platforms. Overall, this work pushes the boundaries of current state-of-the-art in quantifying port hazard—with multi-threaded emulation platform for large-scale (1 million) predictions, built on 300 high-definition simulations on smart unstructured meshes (10 m), using massively parallel multi-GPU-enabled simulations of validated tsunami model VOLNA, and hierarchical file formats—all integrated in an overarching workflow. We illustrate the emulation framework for the Karachi port in the Makran Subduction Zone (MSZ).

The MSZ has given rise to tsunamis in 1524 [19], 1945 [20,21] and 2013 [22]. Recent studies estimate the mega-thrust potential for the eastern part of the MSZ (blue rectangle in figure 1*b*) to be $M_w$ 8.8–9.0 [23]. Thus, here is a pressing need for a comprehensive quantification of tsunami hazard, especially port velocities and associated uncertainties. However, the accurate simulation of tsunami currents at shallow depths requires accurate coastline definition and bathymetry, with adequately refined meshes over a long duration to capture the maximum. Thus, in this study, we employ spatial resolutions of 10 m for the computational mesh, 30 m for bathymetry and 10 m for

coastline, locally in the vicinity of Karachi port, for a total simulation time of 12 h. Furthermore, we employ here an earthquake source designed with segments of size $5 \times 5$ km with carefully constructed positive slip kernels to preserve fidelity to both magnitude scaling [24] and slip scaling relations [25]. The presence of a considerable sediment layer over the MSZ demands incorporation of its influence on the seabed deformation, since an appreciable amplification of up to 60% can be generated [26]. Section 2 describes the models and methods, §3 details the emulation framework, §4 discusses the results and conclusions are drawn in §5.

## 2. Models, data and methods

In this section, we describe in §2a the MSZ, in §2b the finite fault (FF) apparatus and slip profile, in §2c integration of the sediment amplification over the slips for generating seabed deformation (or uplift) and in §2d tsunami propagation.

### (a) MSZ

The MSZ is formed by the subduction of the Arabian plate under the overriding Eurasian plate. It extends approximately 900 km from the Ornach Nal fault (approx. $67°$ E) in the east to the Minab-Zendan-Palami fault (approx. $52°$ E) in the west [20,27,28]. The mega-thrust potential of the entire MSZ is estimated at $M_w$ 9.07–9.22 [23]. Constraints imposed by GPS data resulted in three major segments and an estimated approximately 58% mean coupling ratio between the plates [27]. The subduction interface is divided into the eastern and western MSZ, with the eastern half being more seismically active. Given the scope of this work, we limit ourselves to the eastern MSZ, since tsunamis from western MSZ would have less appreciable effects on Karachi port than those arising from the western MSZ. Furthermore, paleoseismic accounts hypothesize that the western MSZ is seismically inactive compared with the eastern MSZ [29,30].

Here, the probability distribution function (pdf) for the G–R relation is modelled as the doubly truncated exponential distribution [31]

$$G(m) = \begin{cases} \dfrac{\beta e^{-\beta(m-M_w^m)}}{1 - e^{-\beta(M_w^M - M_w^m)}} & M_w^m \le m \le M_w^M \\ 0 & m > M_w^M \end{cases}, \tag{2.1}$$

where $\beta = b \log_e 10$, and the lower $M_w^m$ and upper $M_w^M$ limits of truncation are 4 and 8.8, respectively. The upper limit of $M_w$ 8.8 derives from the mega-thrust potential of eastern MSZ [23]. The rate parameter $b$ of 0.92 is taken from the recent Earthquake Model of Middle East database (see electronic supplementary material, table S2 in [32]), and refers to the whole MSZ. For the scope of this work, we assume it as representative of the eastern MSZ. The complementary cumulative distribution function (ccdf), also called probability of exceedance or survival function, is then:

$$g(m) = \begin{cases} 1 - \dfrac{1 - e^{-\beta(m-M_w^m)}}{1 - e^{-\beta(M_w^M - M_w^m)}} & M_w^m \le m \le M_w^M \\ 0 & m > M_w^M \end{cases}. \tag{2.2}$$

Two cases of the truncated G–R distributions are plotted in figure 2a, i.e. for maximum magnitudes $M_w^M$ of 8.8 and 8.6. Figure 2b shows histograms of actual samples from the distribution (used later in this work).

### (b) Finite fault and slip profile

A FF on the eastern section of MSZ (blue rectangle, figure 1b) is constructed using a total number ($n_F$) of 2295 rectangular segments. The overall dimension of the FF model is $420 \times 129$ km$^2$ ($L^{max} \times W^{max}$). The slip on a segment is denoted by $S_i$, where $i$ varies from 1 to 2295. Okada's closed-form equations transform the slips and other FF parameters into a static vertical displacement denoted by $U$ [33]. The final vertical displacement field results from the combined

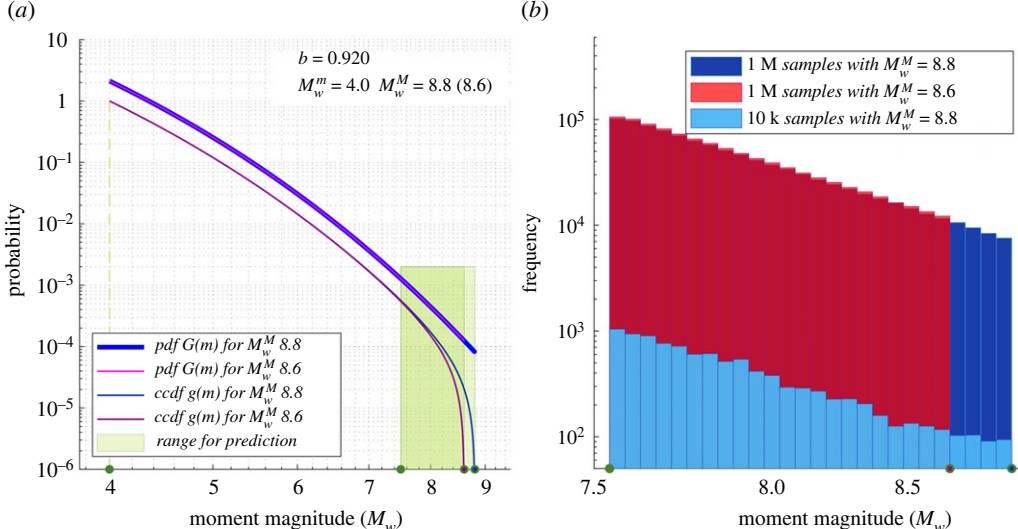

**Figure 2.** Magnitude-frequency distribution for the Makran Subduction Zone (MSZ). (*a*) The Gutenberg–Richter (G–R) relation, showing probability and complementary cumulative distribution functions for two maximum moment magnitude $M_w^M$ assumptions, viz. 8.6 and 8.8. (*b*) Histograms of 1 million (and 10 000) samples of $M_w$. (Online version in colour.)

superposition of vertical displacements due to all the activated fault segments. Among the FF parameters, the dip angle and fault depth ($d_f$) are sourced from the recent plate boundary model, Slab2 [34,35]. The strike and rake angles are kept constant at 270° and 90°.

A segment size ($h_s^2$) is approximately $5 \times 5\,\text{km}^2$ ($l_i \times w_i$), and the segments are arrayed in an $85 \times 27$ grid. This segment size is not chosen arbitrarily. It is selected based on a numerical study of the fidelity of the segmentation viz. $5 \times 5\,\text{km}^2$, $10 \times 10\,\text{km}^2$ and $20 \times 20\,\text{km}^2$ (figure 15*a*) to the earthquake dimension-magnitude scaling relation [24] (figure 1*c*). The discrepancy to the scaling relation appears as discontinuities in the realizable fault lengths ($L$) and widths ($W$) (figure 15*a*, inset). The size of the discontinuities are $\sim h_s$.

We use the definitions of the seismic moment $M_0 = \sum_{i=1}^{n_F} \mu l_i w_i S_i$ and moment magnitude $M_w = (2/3)(\log_{10} M_0 - 9.1)$, with $\mu = 3 \times 10^{10}\,\text{N}\,\text{m}^{-2}$ being the modulus of rigidity. Our implementation of the Okada suite is adapted from the dMODELS[1] code [36,37]. Slips are usually modelled to be uniform on the FF segments, even though inversions of seismic sources evidence localized concentrations of high slips (asperities) over a backdrop of lower slips [12]. Appendix A details the construction of the non-uniform slip profile used in this work.

## (c) Influence of sediment amplification on seabed deformation

Incorporation of the effect of sediments influences tsunami modelling mainly in two ways. First, the interplay of sediment transport and tsunami flow gives rise to enhanced coupled morph- and hydro-dynamics [38,39]. Second, the Okada deformation model [33], with the assumptions of an elastic, homogeneous, isotropic medium in a semi-infinite domain, can be improved by sediment models that exhibit nonlinear, non-homogeneous and an-isotropic behaviour. Considerable amplification (up to 60% locally) of crustal deformation due to the presence of layers of sediments on the seafloor can occur [26]. In this section, we limit the incorporation of the effect of sediments to the deformation model by making use of a sediment amplification curve (figure 3*c*), extracted from elastodynamic simulations of layered sediment-rock seabed [26]. The curve uses the relative

---

[1]v. 1.0 available at pubs.usgs.gov/tm/13/b1/.

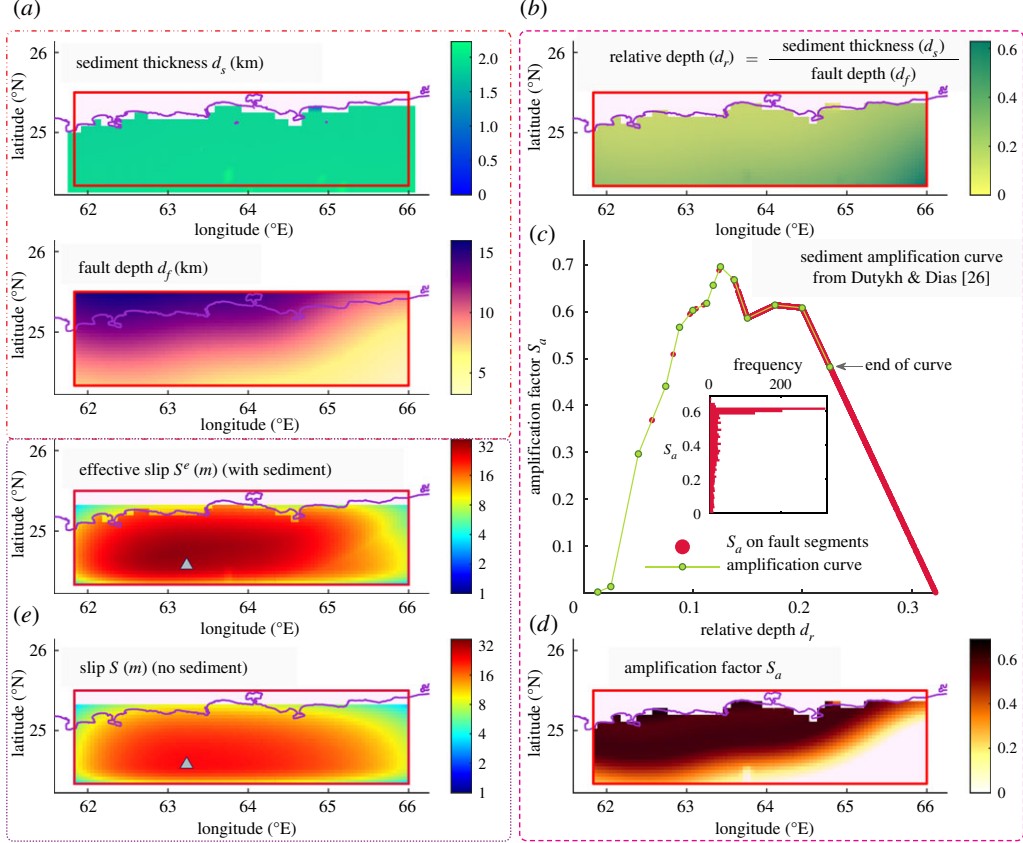

**Figure 3.** Sediment amplification. (*a*) Sediment thickness $d_s$ and fault depth $d_f$. (*b*) Relative depth $d_r$. (*c*) Sediment amplification curve. Inset histogram shows distribution of $S_a$ over the FF segments. (*d*) Sediment amplification factor $S_a$. (*e*) Slip profile $S$ (no sediments), and effective slip profile $S^e$ incorporating influence of sediments through $S_a$. (Online version in colour.)

depth ($d_r^i$) of the *i*th segment (figure 3*b*) calculated as

$$d_r^i = \frac{d_s^i}{d_f^i}, \tag{2.3}$$

where $d_s^i$ is the sediment thickness over the segment interpolated from GlobSed[2] [40], and $d_f^i$ is the down-dip fault depth of the segment taken from Slab2 [34] (figure 3*a*). Given $d_r^i$, the sediment amplification curve supplies the sediment amplification factor ($S_a^i$) on the segment (figure 3*d*). The amplification due to the sediments is incorporated by multiplying the slip $S^i$ with the sediment amplification factor $S_a^i$ resulting in an effective slip $S_i^e$ (figure 3*e*)

$$S_i^e = S_i(1 + S_a^i). \tag{2.4}$$

Okada's closed-form equations transform the effective slips $S_i^e$ into the effective vertical displacement $U^e$ (figure 4*b*) [33]. The influence of sediments not only increases the slips effectively but also modifies the profile, as evident in the emergence of a double-lobed profile (figure 3*e*). The effect is more conspicuous in the associated deformations (compare figure 4*a*,*b*). The amplification factor ($S_a$) peaks at a relative depth of approximately 0.13, after which it decreases. Given the geometry of the fault and overlying sediment profile, a significant number of segments have an amplification factor between 0.4 and 0.6 (or, equivalently 40–60% amplification) (figure 3*c*

[2] Available from ngdc.noaa.gov/mgg/sedthick/.

inset and *d*). Furthermore, the sediment amplification factor is strongly dominated by the fault depth rather than the sediment thickness, which is near-uniform. The sediment amplification curve is defined only till a relative depth of 0.23 [26]. We linearly extrapolate the curve in order to be as conservative as possible in the region where it is not defined as well as to smoothly transition from regions of higher to lower fault depths. The counterparts of average slip $S_{avg}$ and maximum slip $S_{max}$ of $S$ (without sediments) are defined as average effective slip $S_{avg}^e$ and maximum effective slip $S_{max}^e$ of $S^e$ (with sediments). Similarly, effective moment magnitude $M_w^e$ is defined, by replacing $S_i$ with $S_i^e$ in the expression of $M_w$. The effect of sediments on slips are compared in figure 5*a*. Here, the increased scatter of $S_{max}^e$ compared with $S_{avg}^e$ is due to the spatial distribution of $S_a$, which significantly amplifies $S_{max}^e$ depending on the epicentre $(X_o, Y_o)$. Also, the increase in scatter of $S_{max}^e$ as $M_w$ decreases is due to the decrease in fault dimensions that allow many earthquake scenarios to be situated in areas of lower $S_a$. This aspect is pronounced in a similar comparison of $M_w^e$ with $M_w$ in figure 5*b*.

## (d) Tsunami propagation

Analysing wave heights requires few hours of simulation, while investigating the velocities needs longer simulation times. Thus, each scenario is run for 12 h of simulation time $T_s$ to obtain the maximum tsunami velocity and wave height, and is therefore computationally expensive. It is not only imperative that the numerical algorithms in the computer code for tsunami simulations run efficiently at fine mesh resolutions (10 m) needed to capture the currents, but also that the code is amenable to adequate parallelization, e.g. [41,42]. Thus, to run 300 such scenarios, we employ VOLNA-OP2[3] that runs efficiently for unstructured meshes on parallel GPUs [18]. The number of full-fledged scenarios (i.e. 300) is considerably higher than in existing studies related to MSZ [43–45]. Usual simulations employ the Green's functions approach to superpose the tsunami wave heights from a multi-segment FF source. Here, the nonlinear shallow water equations model not only the propagation of the tsunami but also the run-up/down process at the coast [46]. The finite volume (FV) cell-centred method for tessellation of control volume is used in VOLNA, and the barycentres of the cells are associated with the degrees of freedom. Details of numerical implementation, validation against standard benchmarks and comprehensive error analysis are available [18,47]. An important factor affecting the fidelity of long-lasting simulation of currents is numerical dissipation. Giles *et al.* [48] studied the numerical errors in VOLNA-OP2, wherein they are analysed by decomposing them into dispersion and dissipation components. Furthermore, an inter-model benchmarking of different numerical models highlighted the pitfalls in high-resolution current simulations [6]. In line with the scope of this work, we limit our numerical studies using VOLNA-OP2. It may be noted that although the emulation framework is independent of the specific numerical model employed, the accuracy of the emulator is limited by the accuracy of the underlying numerical model. VOLNA models the tsunami life cycle with

$$\frac{\delta H}{\delta t} + \nabla \cdot (H\boldsymbol{v}) = 0 \tag{2.5}$$

and

$$\frac{\delta H\boldsymbol{v}}{\delta t} + \nabla \cdot \left(H\boldsymbol{v} \otimes \boldsymbol{v} + \frac{g}{2}H^2\mathbf{I}_2\right) = gH\nabla b, \tag{2.6}$$

where $H(\boldsymbol{x}, t) = b + \eta$ is the total water depth defined as the sum of free surface elevation $\eta(\boldsymbol{x}, t)$, and time-dependent bathymetry $b(\boldsymbol{x}, t)$. The two horizontal components of the depth-averaged fluid velocity are in $\boldsymbol{v}(\boldsymbol{x}, t)$, $g$ is the standard gravity and $\mathbf{I}_2$ is the $2 \times 2$ identity matrix. The maximum tsunami velocity $v_{max}$ and wave height $\eta_{max}$ at location $\boldsymbol{x}$ at time $t$ are computed as

$$v_{max}(\boldsymbol{x}) = \max_{0 < t \leq T_s} \|\boldsymbol{v}(\boldsymbol{x}, t)\|_2 \tag{2.7}$$

---

[3]v. 1.5 available at github.com/reguly/volna, with improvements to second-order FV scheme and boundary conditions.

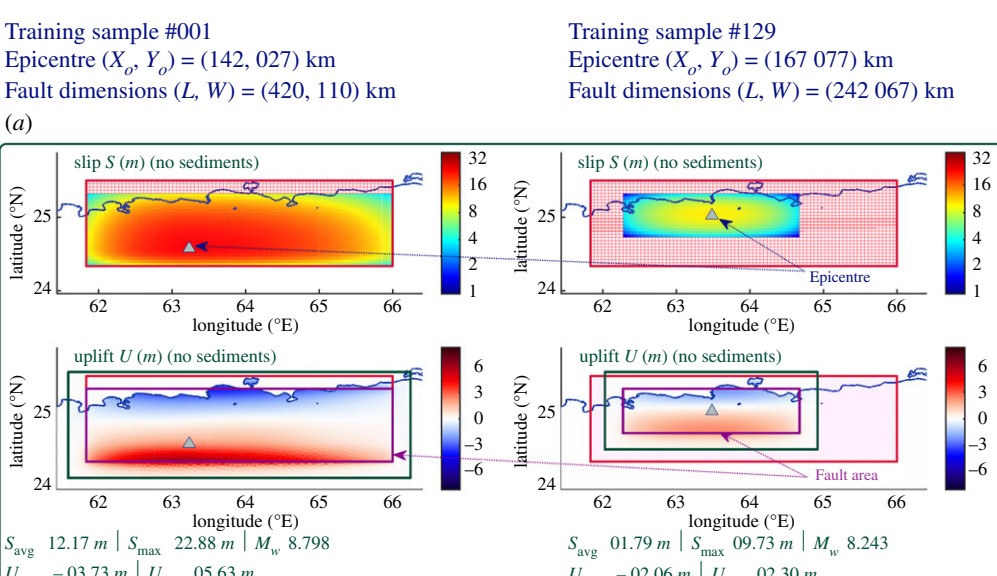

Training sample #001
Epicentre $(X_o, Y_o) = (142, 027)$ km
Fault dimensions $(L, W) = (420, 110)$ km

$S_{avg}$  12.17 m $\mid$ $S_{max}$  22.88 m $\mid$ $M_w$  8.798
$U_{min}$  − 03.73 m $\mid$ $U_{max}$  05.63 m

Training sample #129
Epicentre $(X_o, Y_o) = (167\ 077)$ km
Fault dimensions $(L, W) = (242\ 067)$ km

$S_{avg}$  01.79 m $\mid$ $S_{max}$  09.73 m $\mid$ $M_w$  8.243
$U_{min}$  − 02.06 m $\mid$ $U_{max}$  02.30 m

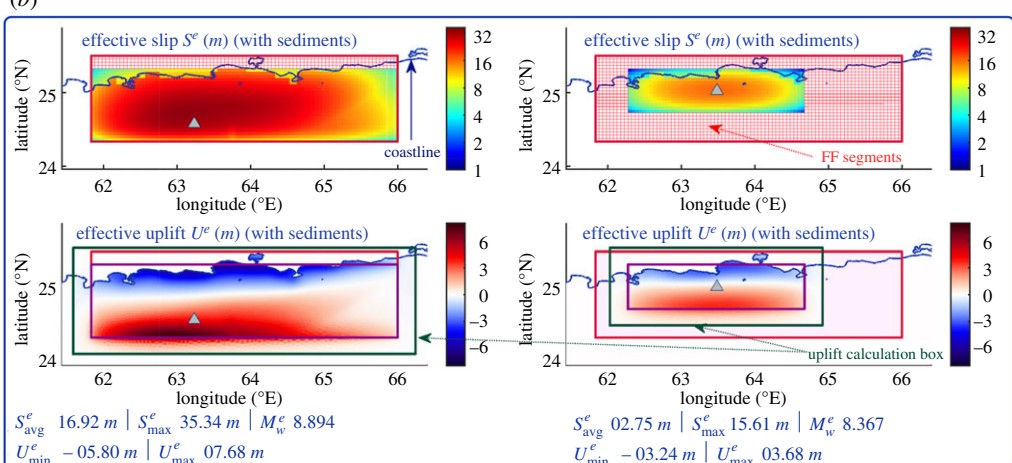

$S^e_{avg}$  16.92 m $\mid$ $S^e_{max}$  35.34 m $\mid$ $M^e_w$  8.894
$U^e_{min}$  − 05.80 m $\mid$ $U^e_{max}$  07.68 m

$S^e_{avg}$  02.75 m $\mid$ $S^e_{max}$  15.61 m $\mid$ $M^e_w$  8.367
$U^e_{min}$  − 03.24 m $\mid$ $U^e_{max}$  03.68 m

**Figure 4.** Comparison of slip and deformation profiles for sample nos. 1 (left column) and 129 (right column). (*a*) Slip *S* and uplift *U* before incorporation of sediment influence. (*b*) Effective slip $S^e$ and uplift $U^e$ with sediment influence. The colourbar for slip is in $\log_2$ scale. (See animations in the electronic supplementary material for a detailed graphical overview of the 300 samples.) (Online version in colour.)

and

$$\eta_{\max}(x) = \max_{0 < t \le T_s} \eta(x, t). \tag{2.8}$$

The dynamic bathymetry $b(x, t)$ is the sum of static bathymetry $b_s(x)$ and $U^e$, the effective deformation due to the influence of sediments. Here, an instantaneous fault is assumed, i.e. $U^e$ is supplied once at the beginning of the simulation. Furthermore, to reduce the computational burden of calculating deformations from 300 events, $U^e$ is computed only within a uplift calculation box covering the fault (green box in figure 4).

Accurate bathymetry, precise coastline and good quality computational mesh are vital for a proper modelling of velocities and currents in shallow water and near the coast. Thus, $b_s$ uses GEBCO 2019 (15″ resolution) [49] complemented with hydrographic charts for Karachi port (approx. 30 m resolution), and SRTM v3 topography (1″ resolution) [50]. For delineating

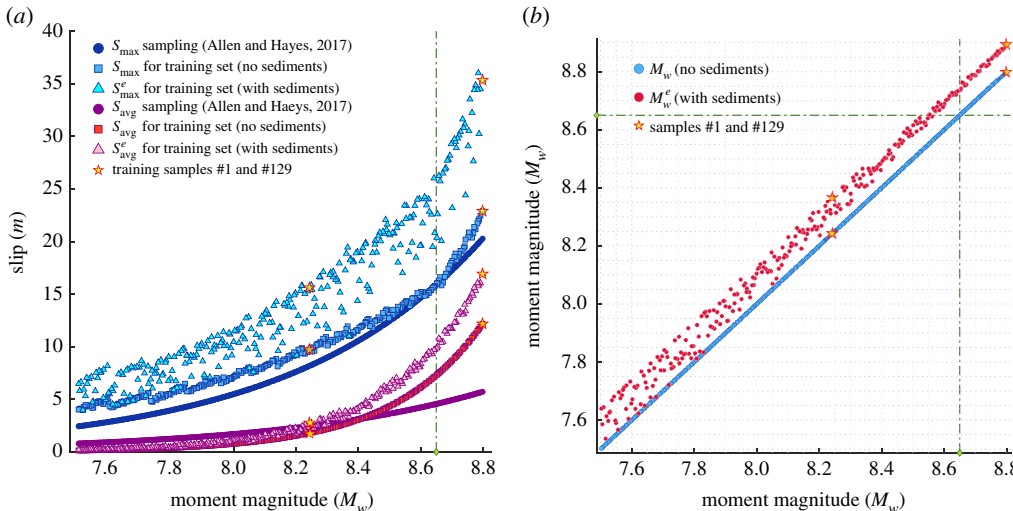

**Figure 5.** Effective slip and moment magnitude. (*a*) Comparison of average and maximum slips with ($S_{avg}^e$ $S_{max}^e$) and without ($S_{avg}$ $S_{max}$) the influence of sediments for the 300 scenarios. (*b*) Same as (*a*) but for moment magnitude. (Online version in colour.)

port structures and breakwaters along the coastline, Google Earth's satellite imagery (approx. 10 m resolution) is used. The merging is described in the electronic supplementary material. The non-uniform unstructured mesh is designed in three stages corresponding to three regions, viz. offshore, onshore and near the port. This three-pronged strategy strikes a balance between having a fine mesh resolution (10 m) near Karachi port and reducing the overall computational cost with approximately $2.64 \times 10^6$ triangles in total. The mesh is generated using Gmsh[4] [51]. The construction of the mesh is described in appendix B.

The outputs $v_{max}$ and $\eta_{max}$ for two training samples (nos. 1 and 129) are plotted in figures 6 and 7, respectively, alongside snapshots taken at various time instants during the simulation.

## 3. Statistical emulation

In this section, emulator training (§3a), diagnostics (§3b) and predictions (§3c) for 1 million events are described.

## (a) Emulator construction

The numerical simulation of the tsunami life cycle, i.e. its generation, propagation and inundation at fine mesh resolutions is computationally expensive due to model nonlinearity, and typically consumes hours on supercomputers. This is all the more prohibitive for a probabilistic quantification where thousands of runs of the tsunami code are required to exhaust the range of plausible scenarios. Statistical surrogates (or emulators) provide a computationally cheap approximation of the complex tsunami solvers, together with estimates of uncertainties in the predictions. In this study, the three input model parameters are moment magnitude ($M_w$) and epicentre coordinates ($X_o, Y_o$) (figure 1*c*, inset). The coordinates have their origin as the southwest corner of the MSZ. The inputs are transformed into effective seafloor deformation. The consequent tsunamis are propagated till Karachi port. The outputs of interest in our case are the maximum wave height ($\eta_{max}$) and maximum wave velocity ($v_{max}$) at $n_G$ (193) virtual gauge locations around the port.

[4]v. 4.4.1 available at gmsh.info.

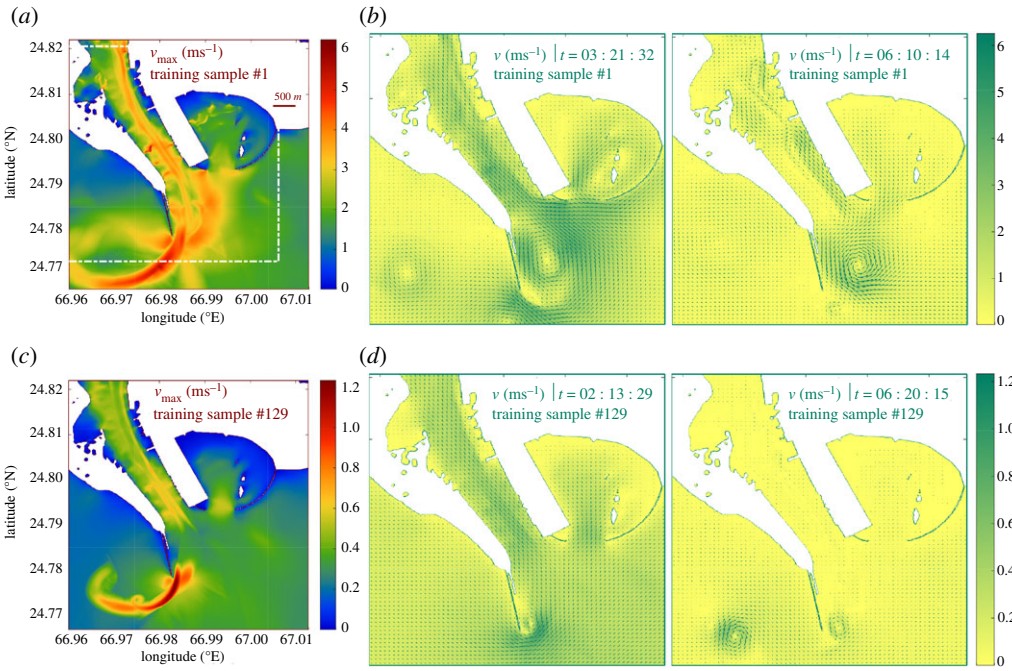

**Figure 6.** Tsunami velocity. (*a*) Maximum velocity at Karachi port over 12 h for sample no. 1. (*b*) Two snapshots of velocities for sample no. 1 restricted to the box (dashed line) in (*a*). (*c,d*) Same as (*a,b*) but for sample no. 129. (Online version in colour.)

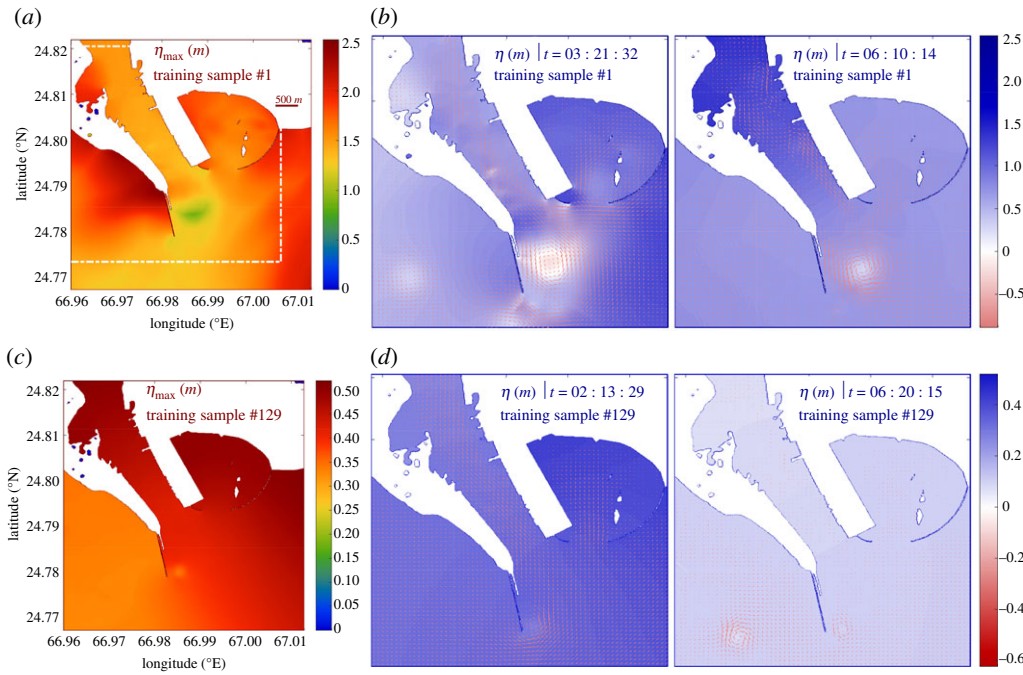

**Figure 7.** Tsunami height. (*a*) Maximum height at Karachi port over 12 h for sample no. 1. (*b*) Two snapshots of heights for sample no. 1 restricted to the box (dashed line) in (*a*). (*c,d*) Same as (*a,b*) but for sample no. 129. (Online version in colour.)

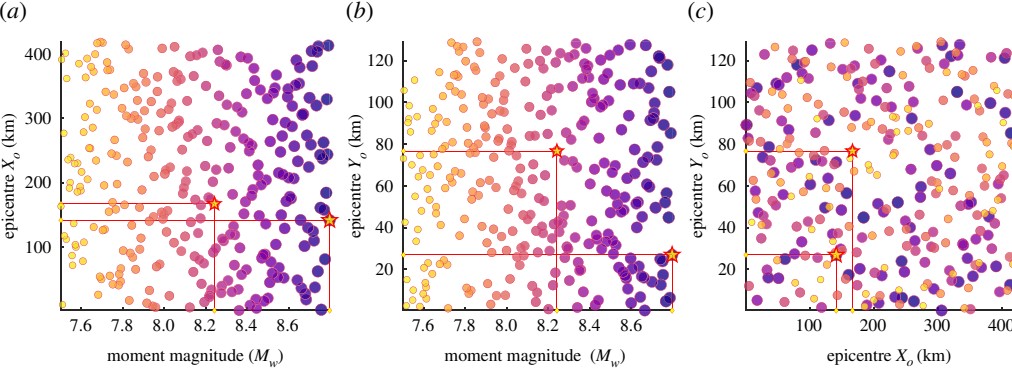

**Figure 8.** Three hundred training scenarios of input parameters ($M_w$, $X_o$, $Y_o$) generated by Latin hypercube design: projections on (a) $M_w - X_o$, (b) $M_w - Y_o$ and (c) $X_o - Y_o$ planes. Sample nos. 1 and 129 are marked with stars. (Online version in colour.)

Thus, the computer code (denoted by $\mathbb{M}$) simulates a multi-physics two-stage physical model, i.e. from the input parameters ($M_w$, $X_o$, $Y_o$) to deformation $U^e$, then from $U^e$ to tsunami outputs $v_{max}$ and $\eta_{max}$. An essential stage is the creation of an informative dataset for constructing the emulator. This is also called the design of computer experiments and the dataset is termed as the training set. The specific purpose of the design stage is to capture the functional relationship between the input parameters ($M_w$, $X_o$, $Y_o$) and output quantities ($\eta_{max}$, $v_{max}$) at a location. The Latin Hypercube Design (LHD) generates a set of points that are nearly uniformly spread to cover the input parameter space. Specifically, it maximizes the minimum distance between points in the set, a feature that explores the functional relationship better than a random scatter. In a physical sense, this spread of points endeavours to capture the information inherent in the input–output relationship as much as possible. The model is evaluated by computer runs of $\mathbb{M}$ at the training points. Here, we employ an LHD of size 300 for three parameters (figure 8). This is large enough to capture complex nonlinear combined sensitivities to the input parameters (e.g. the influence of size and location in relatively small and mid-size events closer to Karachi, or large regional variations in spatial distributions of slips), but still fits within our computational budget. The GP emulator (denoted by $\mathcal{M}$) interpolates across the input–output points in the training set. In other words, the constructed emulator works as an approximation of $\mathbb{M}$, and can be used to generate predictions (or, evaluated) at any point in the space of input parameters. The predictions will be exact at the training points, but uncertain elsewhere. This uncertainty is modelled by a normal distribution whose mean and standard deviation are calculated using the Kriging formula (mean quantities denoted by $\bar{v}_{max}$ and $\bar{\eta}_{max}$) explicitly accounting for the design. This structure allows for any nonlinear relationship to be modelled with uncertainties dependent on the location of the design points, unlike in more standard linear or even nonlinear regressions where the structure is fixed *a priori*. Derivations and exact equations can be found in Beck & Guillas [52]. GP emulation has been instrumental in successfully quantifying uncertainties in tsunami heights generated by landslides over the North Atlantic and the Western Indian Ocean as well as earthquakes over Cascadia [13,53–55]. We use the efficiently implemented multiple-output Gaussian process emulator (MOGP)[5] for emulation.

The covariance kernel is a key component in the construction of the emulator. Here, we use the Matern 5/2 kernel that is smooth enough to avoid a rough GP, but not extremely smooth thus being suitable for modelling the physics. The piecewise polynomial, rational quadratic, exponential and squared exponential functions are other candidates [56]. The parameters (or length scales) in the kernels and other hyperparameters are found via nonlinear optimization

---

[5]v. 0.2.0 available at github.com/alan-turing-institute/mogp_emulator.

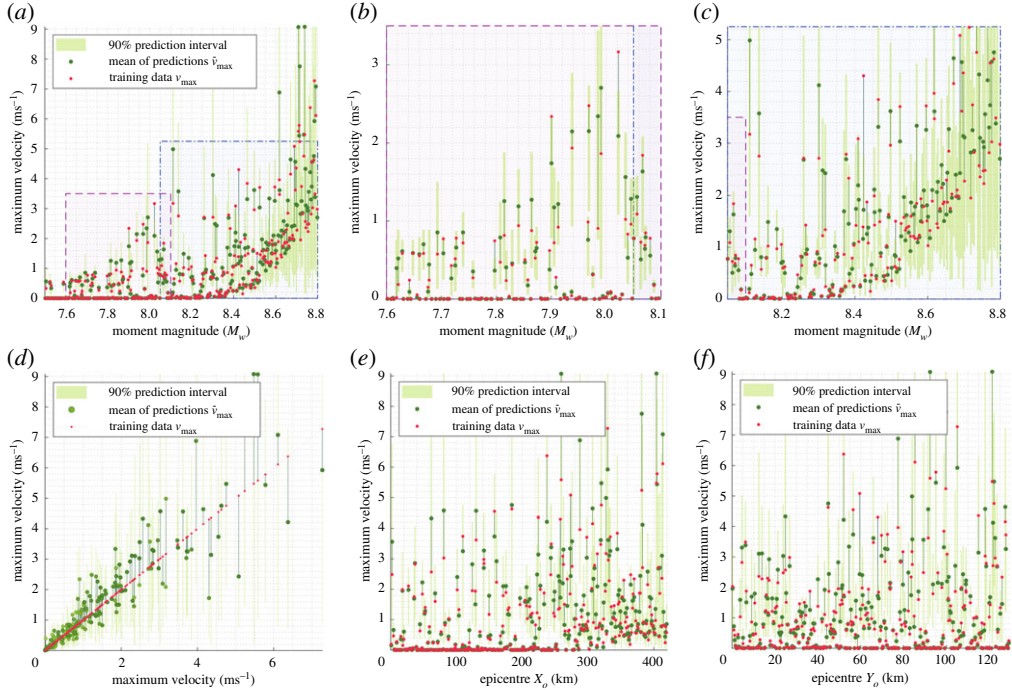

**Figure 9.** Emulator diagnostics (maximum velocity). (*a*) L-O-O data for emulation of maximum velocity $v_{max}$ at a gauge in Karachi port (gauge no. 91). The vertical line segments connect the training data to its predicted counterpart. (*b*) Enlargement of lower moment magnitude region in (*a*). (*c*) Enlargement of higher moment magnitude region in (*a*). (*d*) Data in (*a*) on predicted $\bar{v}_{max}$—training $v_{max}$ axes. (*e*) Data in (*a*) on $X_o$-axis. (*f*) Data in (*a*) on $Y_o$-axis. (Online version in colour.)

(L-BFGS-B) using maximum-likelihood estimation. MOGP also entertains Bayesian approaches as well as a selection of optimization algorithms.

Maximum velocity magnitudes (and heights) are positive. In order to respect this physical constraint and not predict negative velocities (and heights), we feed the logarithm of $v_{max}$ (and $\eta_{max}$) into the construction of the emulator. Since the constructed emulator is now in the logarithmic scale, we transform the predicted quantities back to the original scale by accounting for the lognormal nature of the predicted distributions. Hence, the confidence intervals for the predictions, representing uncertainties, are all rendered positive, and naturally skewed in that direction. Once the emulator is constructed, it needs to be validated before employing it for predictions.

## (b) Emulator diagnostics

In order to validate the quality of the emulation, we provide Leave-one-out (L-O-O) diagnostics here. Our training set consists of 300 pairs of input–output quantities. In L-O-O, a reduced training set of 299 pairs is employed to build an emulator, which is then used to predict the output at inputs of the one pair that was left out. The predicted output (and its uncertainty) is compared with the actual output of the left out pair. This procedure is repeated 300 times to cover all the pairs in the training set. These tests are passed by the emulator, as seen for predicted $\bar{v}_{max}$ in figure 9 and $\bar{\eta}_{max}$ in figure 10. The comparison between the mean of predictions from the emulator $\mathcal{M}$ and the training data from the tsunami simulator $\mathbb{M}$ shows that the emulator approximates well the simulator. The vertical line segments connect the predicted mean with its counterpart in the training data. More importantly, the uncertainties in the predicted mean, quantified in the form of 90% prediction intervals (green bars in figures 9 and 10), represent well the uncertainties about these predictions (or are even slightly conservative), since around

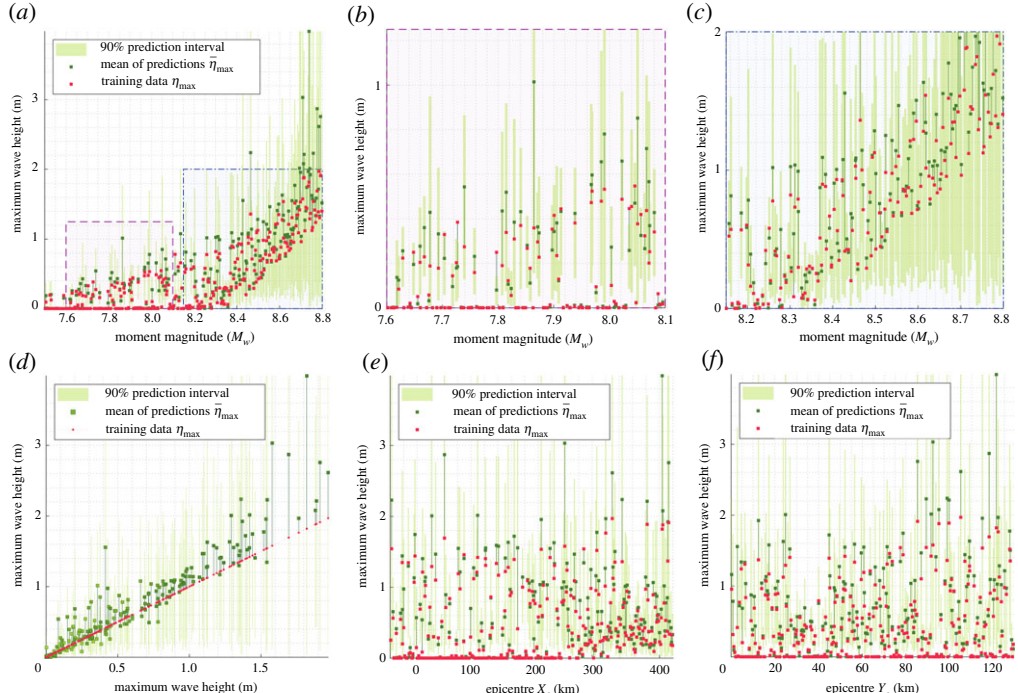

**Figure 10.** Emulator diagnostics (maximum height). (*a*) L–O–O data for emulation of maximum height $\eta_{max}$ at a gauge in Karachi port (gauge no. 91). The vertical line segments connect the training data to its predicted counterpart. (*b*) Enlargement of lower moment magnitude region in (*a*). (*c*) Enlargement of higher moment magnitude region in (*a*). (*d*) Data in (*a*) on predicted $\bar{\eta}_{max}$—training $\eta_{max}$ axes. (*e*) Data in (*a*) on $X_o$-axis. (*f*) Data in (*a*) on $Y_o$-axis. (Online version in colour.)

90% or more of the outputs from the training set fall within these intervals. GP approximation works well inside the convex hull of the training points, but deteriorates near the hull's boundary or exterior giving rise to larger uncertainties in the predictions. For our design, these locations include design limits of $M_w$, and corners or boundaries of the FF, which are limits of $(X_o, Y_o)$. The L-O-O diagnostic indeed shows inadequate fit and larger uncertainties in these regions of the input space. Still, L-O-O provides validation of the emulator inside the convex hull. Furthermore, the L-O-O diagnostics show that some of the lower $M_w$ events do not generate appreciable velocities. In these cases, the location with respect to the port is such that negligible wave energy is radiated to the port. Conversely, the low $M_w$ events that do show appreciable velocities are located such that considerable wave energy reaches the port. The L-O-O also shows a decrease in this positional dependence as $M_w$ increases, due to an accompanying increase in fault area and energy. Additionally, numerical dissipation in the model does play a role here, and numerical schemes tailored for reducing numerical dissipation would increase the accuracy [48].

## (c) Emulator predictions

Although the 300 simulations by themselves generate a good description of the hazard, a large number of scenarios are essential for a comprehensive probabilistic hazard assessment. Thus, we evaluate the model at $n_P$ (1 million) values of $(M_w, X_o, Y_o)$ at 193 virtual offshore gauges. The constructed emulator is used to evaluate the model at inputs that are different from those in the training set. These evaluations are termed predictions. A prediction returns the mean value of the emulated quantity and a measure of inherent statistical error/uncertainty in the approximation, e.g. the standard deviation. Cumulatively, these 193 million predictions not only comprehensively cover the geography around Karachi port but also exhaustively sweep through the range of events

in the magnitude-frequency distribution. Additionally, such a high number of samples is also needed to thoroughly explore the interplay among the three parameters in the input space of $(M_w, X_o, Y_o)$.

The $M_w$ for the 1 million events are obtained by sampling the truncated G–R distribution for the MSZ within our region of interest, i.e. $M_w$ 7.5 to $M_w$ 8.8 (figure 2a). The lower limit of $M_w$ 7.5 is chosen for illustrative purposes. The 1 million values of $(X_o, Y_o)$ are sampled from a uniform distribution defined over the rectangle $[0\ L^{max}] \times [0\ W^{max}]$ of area $420 \times 129\,km^2$. Any changes in the parameters of the G–R relation (i.e. $\beta$, $M_w^m$, $M_w^M$, etc.) only affect the earthquake samples generated for the prediction stage. These changes can be handled in a very efficient manner as the prediction stage is the cheapest component in the entire workflow. In fact, cheap prediction permits fast propagation of uncertainties in the G–R parameters to the hazard intensities. Here, we demonstrate this for two values of one such parameter, the maximum magnitude $M_w^M$. Assuming a reduction of maximum magnitude $M_w^M$ from 8.8 to 8.6 gives a perturbed G–R relation (figure 2a). In this case, the 1 million samples come from the range $M_w$ 7.5 to $M_w$ 8.6. The histograms of 1 million samples for $M_w$ are shown in figure 2b. It also shows 10 000 samples from the range $M_w$ 7.5 to $M_w$ 8.8 for performing comparisons.

To be able to generate 1 million predictions, we employ MOGP. Once the predictions are finished, we are left with two histograms (one each for $\bar{v}_{max}$ and $\bar{\eta}_{max}$) at every virtual gauge, each made up of 1 million samples of predicted quantity. The histograms are processed to extract $P_e(I(x) \geq I_0)$, the probability of exceedance. $P_e$ is the probability of the tsunami having $I(x) \geq I_0$ at a gauge $x$. The intensity $I$ is the measure of hazard, i.e. either $\bar{v}_{max}$ or $\bar{\eta}_{max}$, and $I_0$ is the intensity threshold for the hazard quantity under consideration.

# 4. Results and discussion

We first plot the raw output from the 1 million predictions, i.e. the histograms at 193 gauges in figure 11a,b. At each gauge, two histograms are superimposed on each other. These correspond to the two G–R relations with varying maximum moment magnitude assumptions, i.e. $M_w^M$ 8.6 and $M_w^M$ 8.8 (figure 2). The histograms also act as visual indicators for the measure of the hazard at the gauge, and will be cast as hazard maps in figures 13 and 14. Near the tip of breakwaters and the mouth of the harbour, we observe relatively higher velocities than in other regions. We also observe a complementary relation between the histograms of velocities and wave heights: the gauges having thicker histograms for velocity have thinner histograms for wave heights and vice versa. These phenomena can also be observed in the snapshots (compare figure 6b with 7b).

As expected, there is a clear reduction of hazard when the maximum moment magnitude is reduced. For closer inspection, we enlarge the normalized histograms at gauge no. 91 in figure 11c,d. Gauge no. 91 is located in the centre of the map near the mouth of the port and is chosen since there is substantial spread of both maximum velocities and wave heights in its histograms. In figure 11c, the normalized histograms for maximum velocity are plotted. The range of velocities for $M_w$ 8.8 extends till approximately $16\,ms^{-1}$, while it extends to only approximately $6.2\,ms^{-1}$ for $M_w$ 8.6. Thus, we observe approximately 61% reduction in maximum velocity hazard for a $M_w$ 0.2 reduction in maximum moment magnitude. By comparison, for the same reduction in maximum moment magnitude, the reduction in hazard from maximum wave height is only approximately 38% (from approx. 4.5 to 2.8 m in figure 11d). The probability of exceedance $P_e$ that is extracted from the histograms is plotted in the inset of the respective figure.

Figure 12a,b compare normalized histograms for 1 million (1 M) and 10 000 (10 k) samples of input parameters (figure 2b). The corresponding probability of exceedance $P_e$ plots with their 99% confidence intervals can be seen in the inset. In figure 12a, we observe that the histogram corresponding to 10 000 predictions is curtailed around $7.5\,ms^{-1}$ and becomes very sparse for higher velocities. This is due to a deficit of samples that results in the isolated bars for higher velocities. This behaviour also translates into larger uncertainties (or wider confidence intervals) for estimates of low probabilities of $P_e$. By contrast, 1 million predictions adequately sweep through the entire range of velocities resulting in lower uncertainties (or narrower confidence

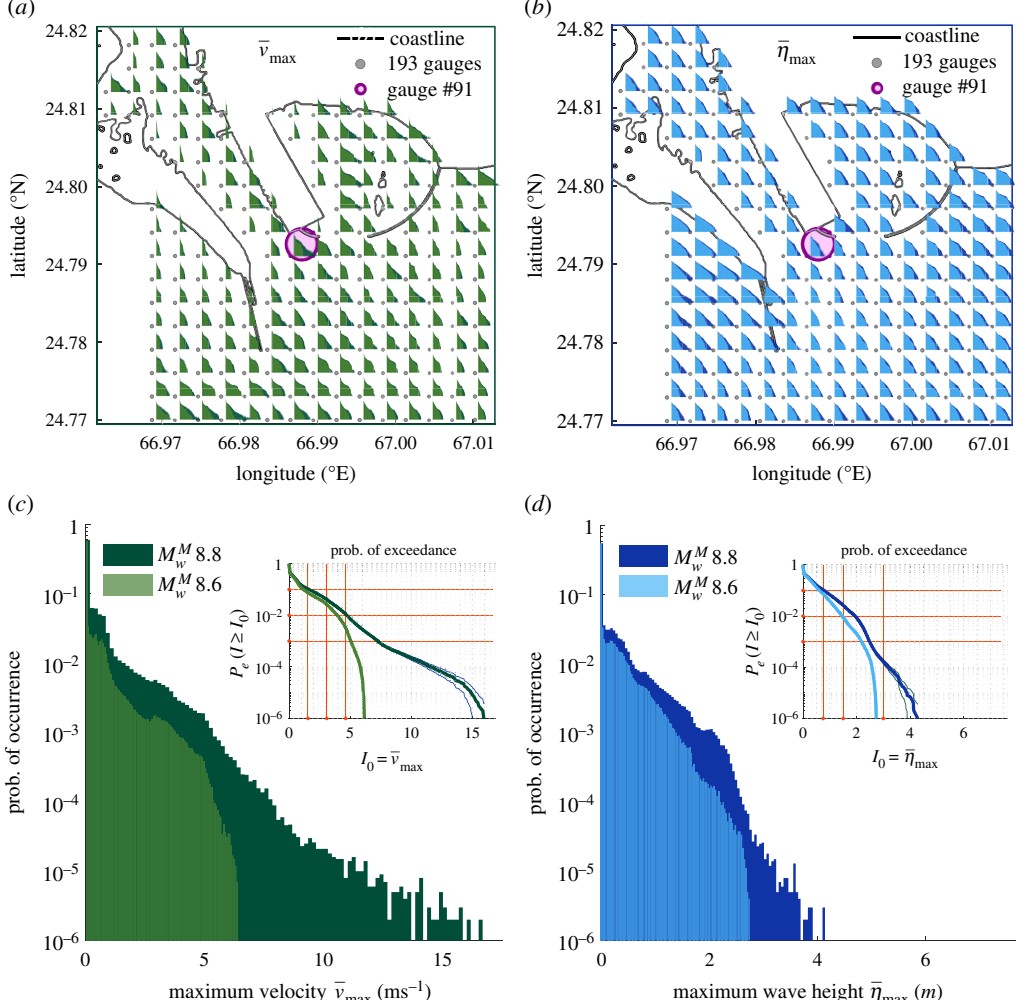

**Figure 11.** 1 million emulator predictions at 193 gauges. (*a*) Histograms of predicted maximum velocities $\bar{v}_{max}$. Histograms from maximum moment magnitude $M_w^M$ of 8.8 and 8.6 are superimposed. (*b*) Same as (*a*) but for predicted maximum heights $\bar{\eta}_{max}$. (*c*) Normalized histograms of $\bar{v}_{max}$ at gauge no. 91. Inset shows probability of exceedance curves, with 99% confidence interval. (*d*) Same as (*c*) but for $\bar{\eta}_{max}$. (Online version in colour.)

intervals) for the tail probabilities. It may be noted that tail probabilities in the $P_e$ curve correspond to extreme events with higher velocities. Similar behaviour is seen in figure 12*b*, where the deficit of samples is observed for maximum wave heights higher than $2.7\,\mathrm{ms}^{-1}$ for the case of 10 000 predictions.

In figure 12*c,d*, we plot the probability of exceedance curves extracted from the histograms of 1 million predictions for the 193 gauges. Superimposed on top are the $P_e$ curves for 10 000 predictions. The horizontal lines in the plots are the chosen values of probability of exceedance, $10^{-1}$, $10^{-2}$ and $10^{-3}$, progressively decreasing by an order of magnitude. The vertical lines in figure 12*c* denote maximum velocities of 1.5, 3.1 and $4.6\,\mathrm{ms}^{-1}$ (or 3, 6 and 9 knots, respectively), values that demarcate categories of damage [5]. The vertical lines in figure 12*d* denote maximum wave heights of 0.75, 1.5 and 3 m. These values are used to construct hazard maps in figures 13 and 14. In both figure 12*c,d*, the reach of the $P_e$ curve is extended beyond the low probability of $10^{-4}$ to include even extreme events only in the case of 1 million predictions. Additionally,

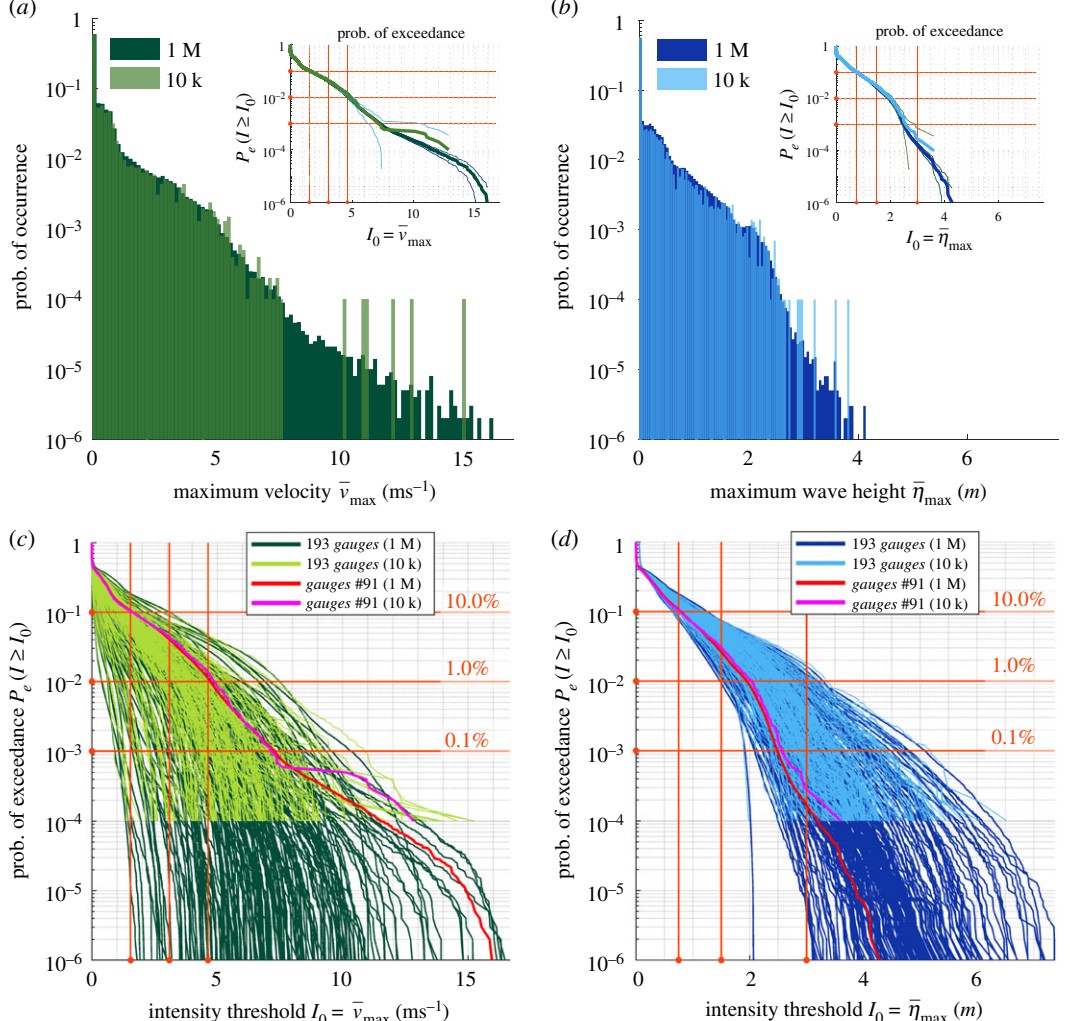

**Figure 12.** Hazard curves. (*a*) Comparison of normalized histograms of 1 million (1 M) and 10 000 (10 k) predicted maximum velocities $\bar{v}_{max}$ at gauge no. 91. Inset shows the probability of exceedance $P_e$ curves, with 99% confidence interval. (*b*) Same as (*a*) but for predicted maximum heights $\bar{\eta}_{max}$. (*c*) $P_e$ curves for $\bar{v}_{max}$ at 193 gauges. Curves from 10 k predictions are superimposed on those from 1 M predictions. Probability and intensity threshold values used to generate hazard maps are shown as horizontal and vertical lines respectively. (*d*) Same as (*c*) but for $\bar{\eta}_{max}$. (Online version in colour.)

although the lower probabilities (around $10^{-4}$) have been made accessible by 10 000 events, they require 1 million events for accurate resolution: with only 10 000 samples, both probabilities and quantities are overestimated between $10^{-3}$ and $10^{-4}$. Hence, being able to produce a very large number of predictions is crucial to hazard assessment. Only with the utilization of the emulator—needing only 300 simulations—are we able to afford realistic predictions of velocities and wave heights at high resolution.

Port hazard is represented on maps by velocity zonations, a time-threshold metric and safe depths for vessel evacuation [3,5]. In this work, the probability of exceedance curves in figure 12 are cast as hazard maps [7,8]. We plot the probability of exceedance at the 193 gauges on the map for the chosen values of maximum velocities in figure 13*a*. Similar plots for chosen values of maximum wave heights are shown in figure 13*b*. For both velocities and wave heights, the overall probability decreases as the intensity threshold increases. Specifically, the bulk of $P_e$ for maximum

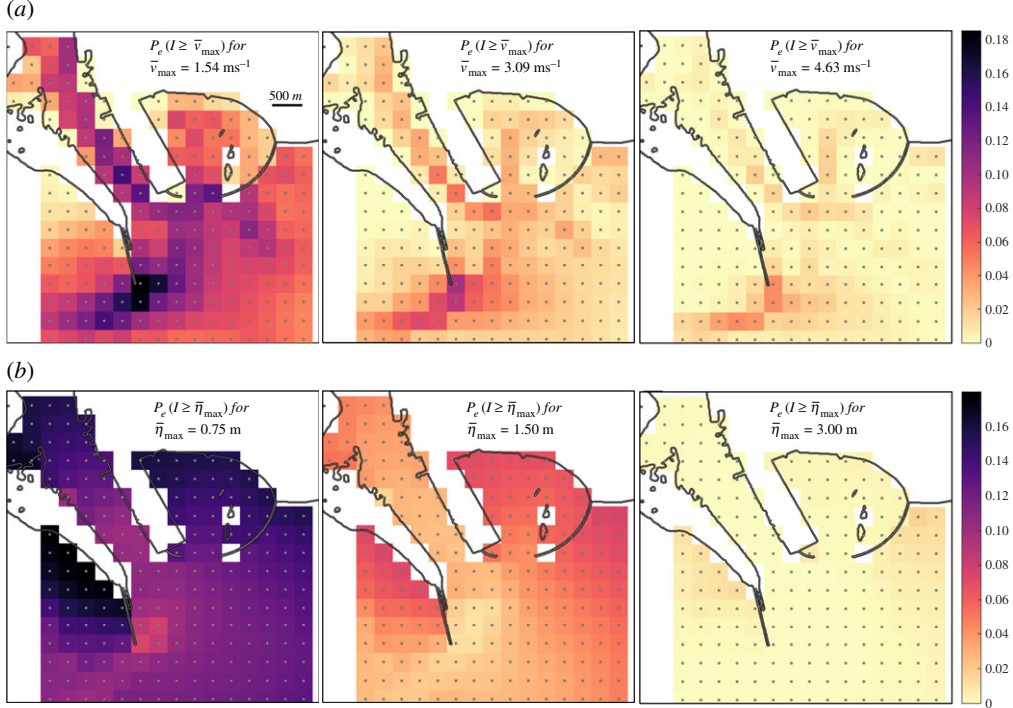

**Figure 13.** Hazard maps. (*a*) Probability of exceedance at 193 gauges for predicted maximum velocities $\bar{v}_{max}$ of 1.5 (left), 3 (centre) and 4.6 ms$^{-1}$ (right). (*b*) Same as (*a*) but for predicted maximum heights $\bar{\eta}_{max}$ of 0.75, 1.5 and 3 m. (Online version in colour.)

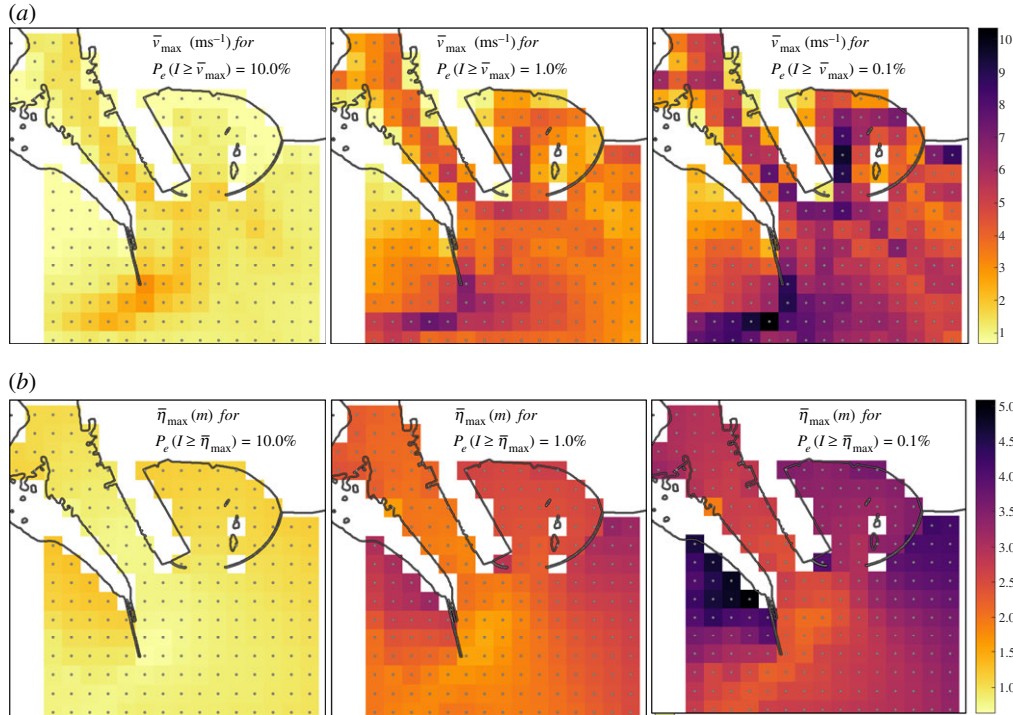

**Figure 14.** Hazard maps. (*a*) Predicted maximum velocities $\bar{v}_{max}$ for exceedance probabilities of $10^{-1}$ (left), $10^{-2}$ (centre) and $10^{-3}$ (right). (*b*) Same as (*a*) but for predicted maximum heights $\bar{\eta}_{max}$. (Online version in colour.)

velocities is concentrated at the tip of breakwaters and along the dredged channel leading into the port (seen in port bathymetry, electronic supplementary material), as also observed in Lynett *et al.* [4]. This is also supported by the patterns of localized higher maximum velocities in figure 6*a*,*c*. By contrast, the spatial distribution of $P_e$ for maximum wave height shows a complementary behaviour and is more spread out.

Conversely, for chosen probabilities of exceedance, the corresponding hazard thresholds at the gauges are plotted in figure 14. As expected, the overall intensity thresholds increase with decrease in probability of exceedance. Again, the bulk of the maximum velocity threshold is concentrated at the tip of breakwaters and along the dredged channel (figure 14*a*). Here too, we see a complementary behaviour for maximum wave height in figure 14*b*.

Velocities have more spatial variation than heights [57], and show increased sensitivity to port configurations, compared with wave heights [58]. The larger spatial variation of velocities in figure 12*c* compared with wave heights in figure 12*d* is evident in the probability of exceedance plotted for all the gauges. This can be attested in figure 11*a*,*b*, where the bulkiness of velocity histograms varies spatially much more than that of the heights. Additionally, at a given gauge, we observe that the spread of velocities is much more than those of the heights for the same set of earthquake scenarios, e.g. compare figure 11*a*,*b* for gauge no. 91. These behaviours can also be observed for individual runs from the spatial variations of maximum velocity and wave height, compare (*a*) and (*c*) in figure 6 to those of figure 7.

The probability of exceedance extracted in this work acts as the basic input for common hazard outputs of probability of occurrence (and return periods), especially the approximately 2475 year mean return period for the maximum considered tsunami as laid out in ch. 6 of ASCE 7-16 [59]. It also feeds into loss estimation functions [60]. Although a full/complete probabilistic description of hazard may remain elusive, a realistic goal of 'fullness' will be to carefully define and perform each step in the PTHA. In these terms, a 'full' probabilistic assessment would ideally need to include further sources of uncertainties, including a thorough analysis of the source uncertainties in its seismic and tectonic setting. These include layers of uncertainties that are either epistemic or aleatoric in nature. Epistemic uncertainties include the scaling relation, and the G–R approximation of the occurrence-magnitude relationship [61], i.e. both the maximum moment magnitude and the *b*-value. For MSZ, the major influence of the maximum magnitude was illustrated in an initial work [62], with a simplified tsunami modelling strategy. Here, we only assess two cases, for $M_w^M$ 8.6 and $M_w^M$ 8.8. Uncertainties in the near shore bathymetry also have a large influence on near shore hazard [63]. Furthermore, the entire MSZ needs to be modelled for an area-wide assessment of hazard at the major ports in Pakistan, Iran, Oman and India, while accounting for crustal, outer-rise and imbricate faults. Secondary tsunamigenic effects from earthquakes in the continental crust (submarine slumps and slides) need additional parameters, e.g. 27 November 1945 $M_w$ 8.1 [64], and 24 September 2013 $M_w$ 7.7 [22] events. Similarly, with appropriate additional parameters, outer-rise and splay faults can be incorporated into the source, e.g. barrier models. Although a large increase in the number of parameters (especially for spatial fields of parameters) presents a challenge to emulation, a solution presents itself in the combination of dimension reduction and emulation [63].

Aleatoric uncertainties in the variations of the geometry in the seafloor uplift and subsidence can be readily incorporated. An alternative to our slip profile generation is to directly parameterize the co-seismic deformation profile using three parameters (or more) [55]. The Okada model that transforms the slips to the vertical deformation is then bypassed. This route is quite attractive since it allows the creation of very realistic deformation patterns with a fixed number of parameters, and does away with the dependency of the deformation/slip on the resolution of the segmentation (shown in figure 15*a*, inset). Our work uniformly samples the 1 million samples for epicentre coordinates (another aleatoric uncertainty). However, a recent spatial distribution of locking has been made available for the MSZ [27]. It would be even more realistic to sample the epicentre coordinates using the locking distribution, since zones of high locking act as a major cause for earthquake recurrence, as recently hypothesized [65]. The locations could be further distributed based on the depth-dependent rigidity [66].

Randomness in tide levels at the time of impact (consequent changes of up to 25% reported [67]) could be included. A better approximation of the currents would be through three-dimensional modelling that accounts for fluid behaviour of the vertical water column and variable vertical flow [6,68]. Better designs of computer experiments than the LHD could be employed to reduce uncertainties in the emulator's approximation, such as sequential design [52]. Instead of investigating a range of scenarios, if one only wants to examine the maximum wave height in order to build defences for instance, a recent surrogate-based optimization could be pursued whereby the design of the experiment is combined with a search for the maximum, saving large quantities of computational time and increasing accuracy due to the focus on the optimization [69]. To be able to emulate a sequence of multiple models of seabed deformation and tsunami propagation, and possibly a three-dimensional model of currents locally, a new approach, called integrated emulation, allows even better designs [70]. The most influential models are run more times where it matters, and the integrated emulator propagates uncertainties with higher fidelity by taking into account the intermediate models in the system of simulators. This approach has the potential to enable fully realistic end-to-end coupling of three-dimensional earthquake sources models with tsunami models [71].

## 5. Conclusion

In this paper, we provide a novel end-to-end quantification of uncertainties of future earthquake-generated tsunami heights and currents in the MSZ:

(i) We replace the complex, expensive high-resolution tsunami simulator by a functionally simple, cheap statistical emulator trained using 300 tsunami simulations at 10 m mesh resolution in the vicinity of the port. We propagate uncertainties from the G–R relation to tsunami impacts of maximum velocities and wave heights in the port area of Karachi, Pakistan. We observe maximum (extreme event) velocities and wave heights of up to $16 \, \mathrm{ms}^{-1}$ and 8 m, respectively, for the range $M_w$ 7.5–8.8 (figure 11).

(ii) We perform the largest emulation using 1 million predictions/source scenarios. To our knowledge, this is the first large-scale uncertainty quantification of earthquake-generated tsunami current hazard. We are able to display the necessity of this very large number of predictions for resolving very low probabilities of exceedance (less than $10^{-3}$)—very high impact extreme events ($v_{\max} > 7.5 \, \mathrm{ms}^{-1}$ and $\eta_{\max} > 3 \, \mathrm{m}$) with tighter uncertainties (figure 12).

(iii) We observe that reduction in hazard due to a reduction in maximum moment magnitude is more for velocities than wave heights. Near the mouth of the harbour, the reduction in hazard is approximately 61% for maximum velocity, but only approximately 38% for maximum wave height (corresponding to a reduction in maximum moment magnitude from 8.8 to 8.6) (figure 12*c*).

(iv) We generate the first area-wide probabilistic hazard maps of tsunami currents from 1 million predicted scenarios at the Karachi port (figures 13*a* and 14*a*). It shows patterns that are geophysically meaningful and important for the next steps of disaster risk reduction. We identify concentrations of high probability of exceedance around the port for given intensity threshold (a maximum of approx. 18%, 10% and 4% for 3, 6 and 9 knots, respectively) (figure 13*a*). Conversely, the same regions also have high intensity thresholds given the probability of exceedance (a maximum of approx. 3.1, 7.5 and $10.3 \, \mathrm{ms}^{-1}$ for 10%, 1% and 0.1%, respectively) (figure 14*a*). Overall, without the large-scale emulation, such outputs would be impractical to produce due to computational costs.

(v) We display more spatial variations for maximum velocity compared with wave heights around the port and their complementary behaviour for the aggregate of 1 million scenarios (figures 6, 7 and 11–14).

**Data accessibility.** The data and codes used have been cited and/or linked in footnotes at first mention.

**Authors' contributions.** M.H. and S.G. conceptualized the problem. S.G. and D.G. conceptualized the employment of large-scale statistical emulation and the inclusion of the effect of sediments. M.H. digitized the bathymetry for Karachi port. D.G. designed the problem with inputs and supervision from S.G. and M.H., developed codes, curated data, carried out the simulations with associated validation, analysis and data processing, and created visualizations for the main article and electronic supplementary material. All authors drafted and critically reviewed the manuscript. All the authors give final approval for publication and agree to be held accountable for the work performed herein.

**Competing interests.** We declare we have no competing interests.

**Funding.** D.G. and S.G. were supported by the Alan Turing Institute under the EPSRC grant no. (EP/N510129/1). M.H. was supported by the Royal Society grant no. (CHL/R1/180173). D.G. was partially funded by the Royal Society-SERB Newton International Fellowship (NF151483). D.G., M.H. and S.G. acknowledge support from the NERC grant no. (NE/P016367/1).

**Acknowledgements.** This work has been performed using resources provided by the Cambridge Tier-2 system (CSD3 Wilkes2) operated by the University of Cambridge Research Computing Service funded by EPSRC Tier-2 capital grant no. EP/P020259/1. The authors would like to acknowledge the use of the University of Oxford Advanced Research Computing (ARC) facility (JADE) in carrying out this work (doi:10.5281/zenodo.22558). Preparative simulations were performed on the EMERALD High Performance Computing facility provided via the EPSRC funded Centre for Innovation (EP/K000144/1 and EP/K000136/1), owned and operated by the e-Infrastructure South Consortium formed by the universities of Bristol, Oxford, Southampton and UCL in partnership with STFC Rutherford Appleton Laboratory. We thank Eric Daub and Oliver Strickson for active development of MOGP UQ suite, Daniel Giles for improvements to second-order FV scheme and boundary conditions in VOLNA-OP2, István Reguly for its installation and running on CSD3, Deyu Ming and Mariya Mamajiwala on truncated G–R distribution and prediction intervals for L-O-O diagnostics, Frédéric Dias on meshing strategies and sediment amplification curve, Theodoros Mathikolonis on emulation, and Simon Day, Kusala Rajendran and C.P. Rajendran on seismicity of M.S.Z. We thank the referees whose detailed comments were instrumental in enhancing the content and presentation of the article.

---

**Algorithm 1** Slip profile generation.

---

1: For a given earthquake moment magnitude $M_w$, find the fault length $L$ and width $W$ from the scaling relation.

2: Fit the fault rectangle of size $L \times W$ into the FF. There are two possibilities, the epicentre $(X_o, Y_o)$ being located: (i) at the centre of the fault and equidistant from the boundaries of the fault rectangle, i.e. with distances $L/2$ and $W/2$, and (ii) away from the centre of the fault. In this case, $(X_o, Y_o)$ is not equidistant from the boundaries of the fault rectangle.

3: Use equation (A 2) to construct the lobes $\phi(x; r_E, \alpha)$ and $\phi(x; r_W, \alpha)$ and form the bi-lobed kernel for fault length $\Phi(x; r_W, r_E, \alpha)$. Similarly, form the bi-lobed kernel for fault width $\Phi(x; r_N, r_S, \alpha)$ by constructing the lobes $\phi(x; r_N, \alpha)$ and $\phi(x; r_S, \alpha)$.

4: Use equation (A 3) to construct the tensor product $\Phi^{\otimes}$ of $\Phi(x; r_W, r_E, \alpha)$ and $\Phi(x; r_N, r_S, \alpha)$.

5: Multiply the values of $\Phi^{\otimes}$ at the centres of each segment (i.e. $\Phi_i^{\otimes}$) with a factor $M_w \left( \sum_{i=1}^{n_F} \mu l_i w_i \Phi_i^{\otimes} \right)^{-1}$ to get the slip $S_i$ on the segment.

---

## Appendix A. Slip profile generation

We fix the dimension $h_s$ of an FF segment based on: (i) computational effort required—scales as $O(n_F) \sim O(h_s^{-2})$ and, (ii) fidelity to the scaling relation (figure 15a, inset)—earthquake dimensions are resolved to $O(h_s)$ (figure 15a). An $h_s \sim 5$ km gives 2295 segments for the overall FF dimensions of $L^{\max} \sim 420$ km and $W^{\max} \sim 129$ km. To resolve the slip profile adequately, we require a fault to span a minimum of four segments along both the length and width directions. Using the scaling relation for $h_s \sim 5$ km, this requirement gives a minimum $M_w$ 6.32 that can be accommodated on the FF. This is sufficient as our region of investigation starts at $M_w^{\min} = 7.5$. The scaling relation also limits the maximum $M_w$ that can be accommodated on the FF area of $L^{\max} \times W^{\max}$, giving $M_w^{\text{sat}} = 8.65$ (figure 1c). Since our region of investigation is till $M_w$ 8.8, for $M_w^{\text{sat}} < M_w < 8.8$, we

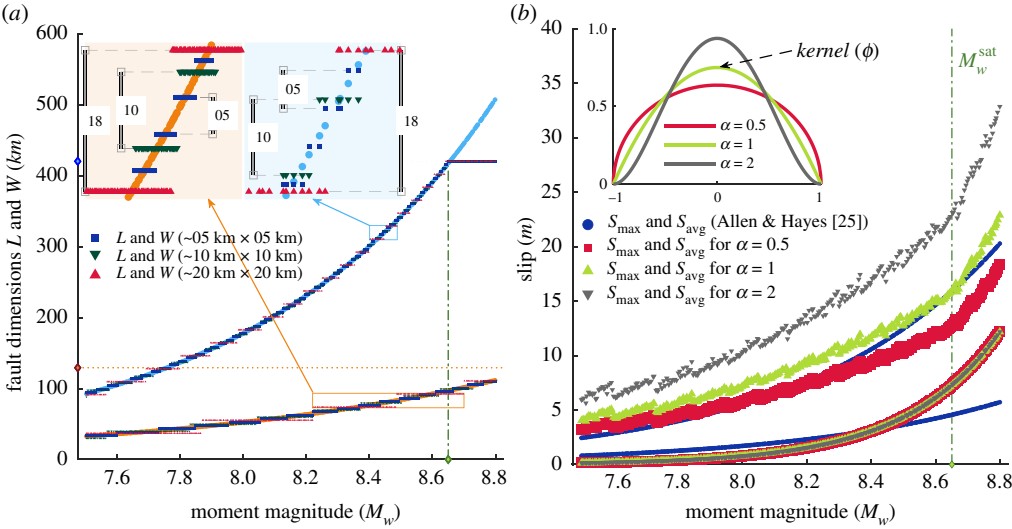

**Figure 15.** (*a*) Realizable fault dimensions made up of approximately 5 km × 5 km, 10 km × 10 km and 20 km × 20 km segments. The inset plots zoom on to the scaling relation to reveal discontinuities. (*b*) Validation of slip profile by varying steepness $\alpha$ and comparing with maximum $S_{max}$ and average $S_{avg}$ slips for 300 scenarios. (Online version in colour.)

proportionately increase the slip on the maximum dimensions. Now, to generate the slip profile, a positive kernel function $\phi$ is used (figure 15b, inset):

$$\phi(x; r, \alpha) = \begin{cases} \dfrac{\Gamma(2\alpha + 2)}{2^{2\alpha+1}\Gamma(\alpha + 1)^2}\left(1 - \left|\dfrac{x}{r}\right|^2\right)^{\alpha} & |x| \leq r \\ 0 & |x| > r \end{cases}, \qquad (A\,1)$$

where the gamma function $\Gamma$ enters the normalization constant, length scale $r$ defines where $\phi$ is non-zero and $\alpha$ adjusts the steepness of $\phi$. With $\phi$ as the core, the bi-lobed kernel $\Phi$ is defined as

$$\Phi(x; r_l, r_r, \alpha) = \begin{cases} \phi(x; r_l, \alpha) & -r_l \leq x \leq 0 \\ \phi(x; r_r, \alpha) & 0 \leq x \leq r_r \end{cases}, \qquad (A\,2)$$

where $r_l$ and $r_r$ are the length scales of the left and right lobes, their values depending on the position of epicentre $(X_o, Y_o)$ with respect to fault length ($L$) and width ($W$). The tensor product of two bi-lobed kernels, one along the length and another along the width of the fault, yields the surface $\Phi^{\otimes}$ (figure 16):

$$\Phi^{\otimes}(x, y; \boldsymbol{r}^{\otimes}, \alpha) = \Phi(x; r_W, r_E, \alpha) \otimes \Phi(y; r_S, r_N, \alpha) \quad (x, y) \in [-r_W, r_E] \times [-r_S, r_N], \qquad (A\,3)$$

where $[-r_W, r_E] \times [-r_S, r_N]$ denotes the domain of the fault and $\boldsymbol{r}^{\otimes} = \{r_W, r_E, r_S, r_N\}$, the distances of western, eastern, southern and northern sides of the fault rectangle from $(X_o, Y_o)$. A normalization of $\Phi^{\otimes}$ with the required moment magnitude yields the final slip profile $S$ (e.g. figure 4). We select $\alpha = 1$ by varying $\alpha$ to mirror the maximum slip $S_{max}$ and average slip $S_{avg}$ curves from empirical scaling relations (see table 2 in [25]) (figure 15b).

## Appendix B. Non-uniform unstructured mesh with local refinement

### (a) Offshore region

The mesh sizing function $h(b_s)$ is based on the merged bathymetry $b_s$ (figure 17a, inset). With the dimensions of the FF earthquake source ($L \times W$), we assume an approximate source wavelength

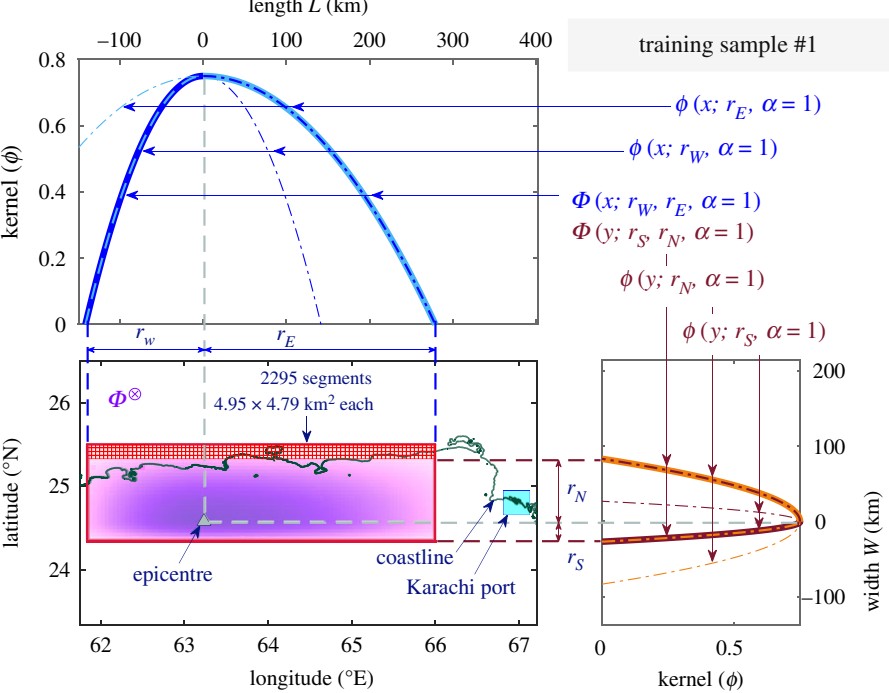

**Figure 16.** Generation of slip profile (for sample no. 1) by tensor product of bi-lobed kernels along fault dimensions. (Online version in colour.)

$\lambda_o$ ($< \sqrt{L^2 + W^2}$) of the tsunami, and a representative ocean depth of the Makran trench $b_o$ (approx. 3 km), to calculate the time period $T_\lambda$ of the wave as

$$T_\lambda = \frac{\lambda_o}{\sqrt{g b_o}}. \tag{B1}$$

Here, $\lambda_o$ is 60 km, which is approximately 60% of the diagonal in the smallest fault, i.e. of size approximately 94 km × 34 km for a $M_w$ 7.5 event (sample no. 300). Assuming the time period to be the same everywhere, the wavelength $\lambda_n$ at depth $b_s(x)$ is found as [73]

$$\frac{\lambda_n}{\sqrt{b_s(x)}} = \frac{\lambda_o}{\sqrt{b_o}}. \tag{B2}$$

This relates the characteristic length of mesh triangle (or mesh size) $h_\lambda(b_s)$ at depth $b_s(x)$ as

$$h_\lambda(b_s) = \left(\frac{\lambda_o}{n_h}\right) \sqrt{\frac{b_s(x)}{b_o}}, \tag{B3}$$

where $n_h = \lambda_n/h_\lambda(b_s) = 10$ is the number of triangles in one wavelength $\lambda_n$. At the coast (i.e. $b_s = 0$), a minimum mesh size $h_m$ (500 m) is specified. The mesh sizing $h_\lambda$ may be steep, or having a high gradient with respect to the bathymetry $b_s$ (green curve, figure 17a). A reduction in gradient is achieved by linearly interpolating the mesh size $\lambda_o/n_h$ at $b_o$ and the minimum mesh size $h_m$ at the coast, i.e. $b_s = 0$ (red curve, figure 17a)

$$h_\mathcal{I}(b_s) = b_s(x) * \frac{(\lambda_o/n_h - h_m)}{(b_o - 0)} + h_m. \tag{B4}$$

The mesh sizing function $h(b_s)$ is then given by the minimum

$$h(b_s) = \min(h_\lambda(b_s), h_\mathcal{I}(b)). \tag{B5}$$

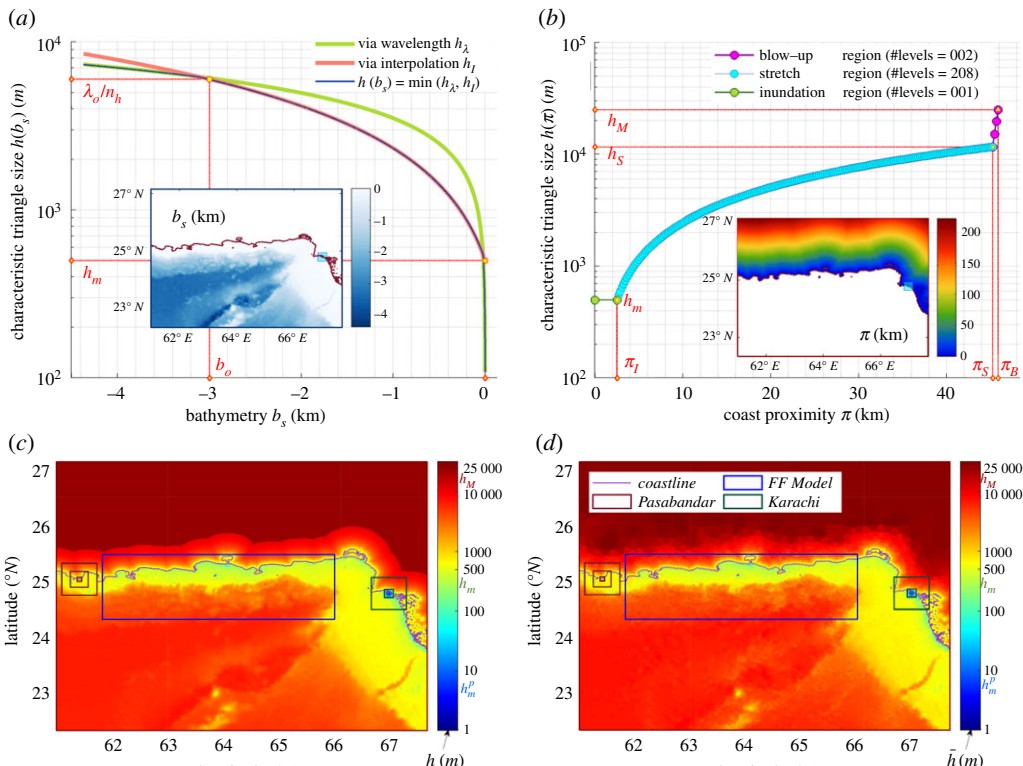

**Figure 17.** Localized non-uniform unstructured mesh. (*a*) Mesh sizing for offshore region based on bathymetry $b_s$ (inset). (*b*) Mesh sizing for onshore region based on coast proximity $\pi$ (inset). (*c*) Mesh sizing function $h$ supplied to Gmsh. (*d*) Mesh sizes $\bar{h}$ in mesh generated by Gmsh using $h$ in (*c*). (Online version in colour.)

## (b) Onshore region

The mesh sizing function $h(\pi)$ is based on the coast proximity $\pi(x)$ (figure 17*b* inset), which is defined as the minimum distance of point $x$ from the coastline $\mathcal{C}$ of the merged bathymetry $b_s$

$$\pi(x) = \min_{x_c \in \mathcal{C}} \|x - x_c\|_2. \tag{B 6}$$

The construction of $h(\pi)$ is split into three regions, *viz.* inundation, stretch and blow-up (figure 17*b*). In the inundation region, which is defined to extend inland for a distance $\pi_I$ (2.5 km) from the coast, the mesh size is prescribed as the minimum mesh size $h_m$ (500 m). Thus, the inundation region facilitates smooth transition between the onshore and offshore meshes. Further inland, we require the triangle sizes to explode quickly to the maximum mesh size $h_M$ (25 km). This region is called the blow-up region (from $\pi_S$ to $\pi_B$ in figure 17*b*). We introduce the stretch region between the end of the inundation region and the beginning of the blow-up region (i.e. from $\pi_I$ to $\pi_S$ in figure 17*b*), for a gradual transition of corresponding mesh sizes, i.e. from $h_m$ to $h_S$ (10 km). This gradual change is achieved by setting the size ratio $\rho$, which is the ratio of characteristic lengths of adjacent triangles (or grading gauge [74]) to 1.3. The stretch distance $\pi_S - \pi_I$ is calculated as

$$\pi_S - \pi_I = h_m + \rho h_m + \rho^2 h_m + \ldots + \rho^{n_S} h_m. \tag{B 7}$$

Equation (B 7) is a geometric series that approximates the distance by summing up the sizes of $n_S + 1$ triangles, lined up end-to-end in a straight line, monotonically increasing in size by a factor

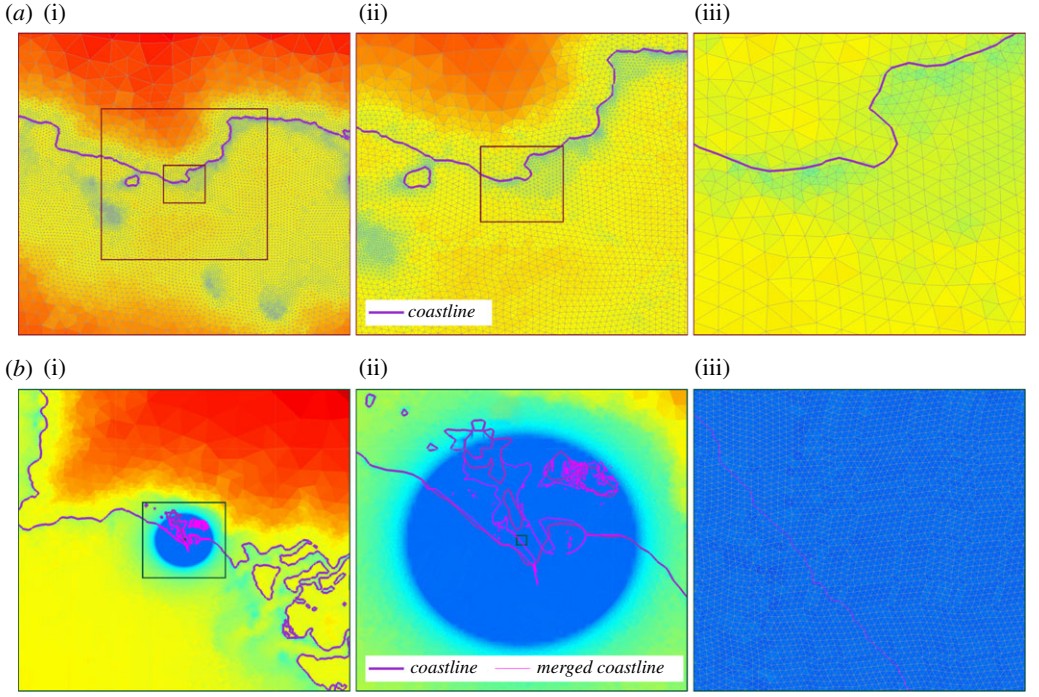

*(a)* (i) (ii) (iii)

— coastline

*(b)* (i) (ii) (iii)

— coastline — merged coastline

**Figure 18.** Localized non-uniform unstructured mesh. (*a*) Mesh without local refinement at Pasabandar shown at scales of (i) 64 km × 64 km, (ii) 32 km × 32 km and (iii) 8 km × 8 km, respectively. (*b*) Locally refined mesh at Karachi port shown at scales of (i) 64 km × 64 km, (ii) 16 km × 16 km and (iii) 0.5 km × 0.5 km, respectively. (Online version in colour.)

of $\rho$ [74], starting from $h_m$ to $\rho^{n_S} h_m$. Equating the last term to $h_S$, solve for integer $n_S$ as

$$n_S = \left\lceil \log_\rho \left( \frac{h_S}{h_m} \right) \right\rceil, \tag{B8}$$

where $\lceil \cdot \rceil$ denotes the ceiling function. Similarly, the blow-up distance $\pi_B - \pi_S$ is calculated as

$$\pi_B - \pi_S = h_S + \rho h_S + \rho^2 h_S + \ldots + \rho^{n_B} h_S. \tag{B9}$$

The description of equation (B9) is similar to equation (B7). Equating the last term to $h_M$, get integer $n_B$ as

$$n_B = \left\lceil \log_\rho \left( \frac{h_M}{h_S} \right) \right\rceil. \tag{B10}$$

Note: The mesh sizing functions $h(b_s)$ and $h(\pi)$ are specified to Gmsh on a background rectangular grid that has half the resolution (approx. 210 m) of GEBCO 2019 grid, sufficient for specifying the minimum mesh size $h_m$ (500 m). The number of levels in figure 17*b* are the number of grid points needed in the background mesh to specify mesh sizes in the respective region.

*Port region*: The strategy is similar to that in the stretch region, but the radial distance from the centre $(x_p, y_p)$ of the DOI (or port) is used instead of the coast proximity. The mesh size is fixed at $h_m^p$ (10 m) in the DOI where resolved bathymetry is available. The resolution of background rectangular grid near the port is $10\,\text{m} \sim h_m^p$. A smaller size ratio $\rho^p$ of 1.05 ensures a gradual transition of mesh sizes. In increasing radii extending outwards from the DOI, the mesh size increases similar to equation (B7), but iteratively with an increasing number of terms in the geometric progression. The iterative procedure is employed to effect a smooth transition of the mesh at the port with existing offshore and onshore meshes (figure 18*b*). For contrast, figure 18*a* shows Pasabandar port, where local refinement of mesh is absent.

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
