## [Peer Review File · Proceedings. Mathematical, Physical, and Engineering Sciences]

Review History

RSPA-2020-0328.R0 (Original submission)

Review form: Referee 1

Is the manuscript an original and important contribution to its field?

Acceptable

Is the paper of sufficient general interest?

Acceptable

Is the overall quality of the paper suitable?

Marginal

Can the paper be shortened without overall detriment to the main message?

No

Do you think some of the material would be more appropriate as an electronic appendix?

Yes

Do you have any ethical concerns with this paper?

No

Recommendation?

Major revision is needed (please make suggestions in comments)

Comments to the Author(s)

This paper introduces a probabilistic approach to assess the tsunami currents hazards in the nearshore based on a statistical method, called an emulator. Such a 'new' type of tool is expected to provide an efficient scheme for analyzing tsunami hazards and to help break through barriers that other PTHA based on numerical modeling may have. Meanwhile, there are major issues to be responded to before this work is published, as listed below.

Main concerns

1. Throughout the entire manuscript, figures weren't presented in an appropriate way, too many windows and information condensed in a single figure slot. This led to lose the focus of each figure. For example, Fig. 5 & 7 can be re-arranged. Figure 6 is not so significant in the main body of the manuscript and can be relocated in either appendix or supporting material if applicable. Fig. 8 and others can be lessened by removing some sub-plots or by merging some of them. So I strongly encourage the author to review their figures and evaluate them to determine which are crucial to be shown in and also separate them into two or more. This will increase the visibility of the included figures and aid the reader to follow them easily.
2. The author can put more stress on Section 2. (e), (f), (g) since they contain a huge part of novelty (i.e., emulation scheme) of the present study. Such an effort will distinguish what is typical and what is new in this research. I recommend the author separate them from Section 2 in some way.
3. Abstract needs to be re-drafted to concisely address the significance of this study. Currently stated are the modeling process, which is not able to deliver and efficiently highlight the main contribution of this study to the tsunami research. Thus, the rewritten abstract should state the main steam of the study, clearly declaring the contribution of the main findings that had been achieved.
4. Introduction is required to be reshaped. For example, a detailed description of what happened at the Salalah port during the 2004 Sumatra-Andaman tsunami can be omitted. Rather than including that, the authors can endeavor to top up more reviews on probabilistic methods since the core part of this study is how to access the tsunami hazards on a statistical basis. Therefore, more literatures stating the probabilistic assessment of tsunami hazards need to be thoroughly and comprehensively reviewed and added (e.g., Zhang and Niu (2020)).
5. It is mentioned that "the strong currents continued for hours after the waves with maximum amplitude had arrived (nearly 9h in Salalah.)" (Page 2 line 31). This long-lasting currents simulation can be achieved only when the higher-order scheme guaranteeing the least numerical diffusion is applied. Therefore, numerical dissipation embedded in the applied model, VOLNA-OP2, which may corrupt the modeling results needs to be justified along with a wide range of reviews on other tsunami models in this regard. Moreover, the uncertainty originating from using a specific type of model should be examined. Lynett et al. (2017) provided the comprehensive simulation results from various numerical models focusing on the tsunami-current generation and inter-compared them. It suggested that the huge uncertainties may result from using different types of numerical models, particularly when a high-definition grid less than 10 m is applied.
6. What is 'rupture origin' Did the authors mean an epicenter by mentioning it? If so, an epicenter must be a better expression. Also, the terms, 'fault' and 'rupture' were being used confusingly in several parts, which is not acceptable in the scientific journal. Please make a clear definition of each ahead of its use and use it appropriately when required.

7. The description of 300 earthquake scenarios is not enough for readers to figure them out. The authors need to detail how the 300 cases had been constructed, through a sort of the table showing the entire combination of themselves. The algorithm to generate 300 scenarios based on Latin Hypercube Design should also have physical sense. Moreover, developing a million predictions from the 300 cases is not well understood despite given explanation and figures, and thus conclusions drawn from those results are not so convincing.

8. The sediment amplification and the resulting bathymetric change introduced by Eq. (2.11) seem to lost lots of physical standard. For example, there must be another set of equations for bathymetric changes coupled to Eq. (2.7) and (2.8) (e.g., Li et al. (2012), Jaffe et al. (2016)). Please address how the authors handled this issue.

9. When the bathymetries from several sources are merged, special care needs to be paid because they may have different datum. Please explain how the author handled this issue.

10. How do the authors define 'high-definition' of resolution in bathy, mesh, and coastline? It seems it refers $O(10m)$, but there have been similar achievements up to those resolutions so far and it is not a quite challenging issue.

11. Figure 2: It is not easy to catch the main flow of Figure 2. Further simplification and conceptualization are required for the elegant representation. Also, the section number included in the figure didn't match with one in the main body.

13. Please cite Yuan et al. (2020) where applicable, since it provided good accomplishment of using GPU in tsunami modeling.

References

Jaffe B, Goto K, Sugawara D, Gelfenbaum G, La Selle S. 2016. Uncertainty in tsunami sediment transport modeling. *Journal of Disaster Research* 11(4): 647-661.

Li L, Qiu Q, Huang Z. 2012. Numerical modeling of the morphological change in Lhok Nga, west Banda Aceh, during the 2004 Indian Ocean tsunami: understanding tsunami deposits using a forward modeling method. *Natural Hazards* 64: 1549-1574.

Lynett, P. J., Gately, K., Wilson, R., Montoya, L., Arcas, D., Aytore, B., ... & David, C. G. (2017). Inter-model analysis of tsunami-induced coastal currents. *Ocean Modelling*, 114, 14-32.

Yuan, Y., Shi, F., Kirby, J. T., & Yu, F. (2020). FUNWAVE-GPU: Multiple-GPU Acceleration of a Boussinesq-Type Wave Model. *Journal of Advances in Modeling Earth Systems*, 12(5), e2019MS001957.

Zhang, X., & Niu, X. (2020). Probabilistic tsunami hazard assessment and its application to southeast coast of Hainan Island from Manila Trench. *Coastal Engineering*, 155, 103596.

Review form: Referee 2

Is the manuscript an original and important contribution to its field?

Good

Is the paper of sufficient general interest?

Good

Is the overall quality of the paper suitable?

Acceptable

Do you have any ethical concerns with this paper?

No

Recommendation?

Major revision is needed (please make suggestions in comments)

Comments to the Author(s)

Review of the manuscript: "Probabilistic Quantification of Tsunami Current Hazard using Statistical Emulation" by D. Gopinthan et al.

The manuscript presents a method to evaluate the probabilistic hazard of tsunami currents in specific target sites (e.g., harbours), using statistical emulators to reduce the computational resources. Moreover, the work is focused on tsunami currents which present some differences in terms of hazard with respect the more common use of tsunami wave heights as hazard intensity measure.

The topic is of interest for the tsunami community and I agree with authors that both the method (statistical emulator) and the kind of intensity measure (tsunami currents) are still not much explored in tsunami science.

However, I have some comments that, in my opinion, should be addressed before the publications.

First of all, I have a very general comment.

My main concern is the possible misleading of the actual goal of the manuscript: in its present form, the work is illustrative of a general workflow that can be used to assess probabilistic tsunami hazard. Each step is technically detailed in its implementation and this is a really worthy effort from the authors. But, on the other hand, I would stress that they should be careful to avoid that a reader could be confused that this could be an example of hazard assessment. To be clearer: immediately in the abstract it is stated that the work "[...] map probabilistic representations of maximum tsunami velocities and heights for the full range of possible future tsunamis at around 200 locations around Karachi port, Makran Subduction Zone (MSZ)". In my opinion, this is not true. A rigorous source analysis from the seismic and tectonic point of view is missing, so I disagree with authors that the "full range" of possibilities is here considered. For this reason, I suggest that in some part of the text (likely in the title itself? "A method to quantify..." or something like this?) authors should try to stress more that this work is purely illustrative of a method.

My thoughts are partially confirmed by the authors themselves: indeed, in the discussion, they are more prudent, with a list of missing sources of uncertainty that should be considered before performing "a full probabilistic assessment" (page 21, lines 51 and following): scaling relations, G-R maximum moment magnitudes, bathymetry. So they are in part aware of this.

Moreover, I think that also the seismic source variability is only partially explored and poorly described; in my opinion, this is an essential component of such hazard analyses and should at least presented. Indeed, even though the case study is illustrative, the MSZ source zone and the Karachi Port are real, so I think that it is important to introduce the geophysical context and better address some input choices.

I'm not asking to explicitly explore the full source variability: but I think that authors should, at least, add an exhaustive section to introduce the tectonic setting of the area and better explain that they are considering only a part of the potential sources for illustrative purposes.
 Here, for example, the authors are excluding the western part of the MSZ subduction, which is already a quite important choice (see my comment #1). Then, what about other potential

tsunamigenic seismic sources? Could splay faults, outer-rise earthquakes and crustal earthquakes generate tsunamis with impact for the Karachi port? Could these sources be easily incorporated in the present procedure or they represent a challenge? Please enrich the text with this part.

Finally, some subjective choices are just presented as they are, without explanations: this could lead a reader to think that these could be common choices, but they aren't (see my comment #2 about the range of magnitudes).

Other comments:

1. Why excluding the western part of MSZ? It seems to me that the illustrative example could be more solid if applied to the whole subduction instead to a part of it. The considered G-R distribution refers to the whole subduction or just to the eastern part? Could you please address this? I wasn't able to find Table S1 (mentioned at page 17, lines 34-35) so I could not check how the authors selected the b-value. However, if this extension to the western MSZ does not present technical issues, why do not model it? The results would be representative of the whole subduction and could be more interesting to comment.

2. Why the minimum considered magnitude is 7.5? for saving numbers because of illustrative purposes or there is any geophysical reason to neglect magnitudes < 7.5? Small tsunamis couldn't generate currents as well? Please address this point.

3. I found that the state of the art is not complete. For example, other works describing general methods to propagate the uncertainty and/or to reduce the number of tsunami simulations required to perform probabilistic tsunami hazards have been published (Lorito et al., 2015, Volpe et al., 2019 and very recently Kotani et al., 2020 could be of interest). I think that, even though it was on landslide-generated tsunami, the work on statistical emulators by Sarri et al. in 2012 represents an existing reference of a similar approach and merit be mentioned.

4. Authors could re-think the manuscript's structure, for example by moving to Appendix some of the most technical parts, in order to lighten the reading of the main text, which is now very dense of figures and subfigures that could distract the reader.

One explicit example is the following: Appendices A, B and C have a corresponding section with the same title also in the main text (Slip profile generation, Merging of Bathymetries and Localised Non-Uniform Unstructured Mesh); I think that this is redundant: I suggest to keep the Appendices only and also move there the corresponding figures.

5. This is just a suggestion for a future implementation: a further module that could be incorporated in the procedure is the effect of the water layer between the seafloor displacement and the corresponding water surface perturbation (see Kajiura, 1963).

Some minor comments:

- I can't find any table mentioned in the text. Maybe there was an issue during files' submission?

- Introduction, page 2, line 24: A $M_w=7.7$ earthquake is not small. I think authors can remove the adjective "small" which is not necessary (and not understandable) in this sentence.

- Introduction, page 2, line 26: I cannot find Table 1 and reference to Figure 3 seems not correct, please check.

- Introduction, page 2, line 28: I suggest to use labels when referring to subplots (ad example Figure 1b) to help the reader.

- Why, in figures 4, 5 and 9, the 3000 m depth contour line is explicitly shown? It seems to me a little confusing with the coastline. Is it necessary in these plots? If not, you could remove it and

keep only the colormap in figure 7a for water depth.

References:

Kajiura K., 1963. The leading wave of a tsunami, *Bull. Earthq. Res. Inst. Univ. Tokyo*, 41, 535–571

Kotani T., K. Tozato, S. Takase, S. Moriguchi, K. Terada, Y. Fukutani, Y. Otake, K. Nojima, M. Sakuraba, Y. Choe, Probabilistic tsunami hazard assessment with simulation-based response surfaces, *Coastal Engineering*, 160, 2020, doi:10.1016/j.coastaleng.2020.103719

Lorito S., Selva, J., Basili, R., Romano, F., Tiberti, M. M., & Piatanesi, A. (2015). Probabilistic hazard for seismically induced tsunamis: Accuracy and feasibility of inundation maps. *Geophysical Journal International*, 200(1), 574–588. <https://doi.org/10.1093/gji/ggu408>

Sarri, A., Guillas, S., and Dias, F.: Statistical emulation of a tsunami model for sensitivity analysis and uncertainty quantification, *Nat. Hazards Earth Syst. Sci.*, 12, 2003–2018, <https://doi.org/10.5194/nhess-12-2003-2012>, 2012

Volpe, M., Lorito, S., Selva, J., Tonini, R., Romano, F., and Brizuela, B.: From regional to local SPTHA: efficient computation of probabilistic tsunami inundation maps addressing near-field sources, *Nat. Hazards Earth Syst. Sci.*, 19, 455–469, <https://doi.org/10.5194/nhess-19-455-2019>, 2019.

Decision letter (RSPA-2020-0328.R0)

04-Sep-2020

Dear Dr Gopinathan:

I am writing to inform you that your manuscript RSPA-2020-0328 entitled "Probabilistic Quantification of Tsunami Current Hazard using Statistical Emulation" has been rejected in its present form for publication in *Proceedings A*.

The Editor has made this decision based on the advice of referees, and taking into account their own opinion of your paper. With this in mind we would like to invite a resubmission, provided the comments of the referees and any comments from the Editor are taken into account. This is not a provisional acceptance.

The resubmission will be treated as a new manuscript. Please note that resubmissions must be submitted within six months of the date of this email. In exceptional circumstances, extensions may be possible if agreed with the Editorial Office.

Please find below the comments made by the referees, not including confidential reports to the Editor, which I hope you will find useful. If you do choose to resubmit your manuscript, please include details of how you have responded to the comments, and the adjustments you have made.

Please note that we have a strict upper limit of 28 pages for each paper. Please endeavour to incorporate any revisions while keeping the paper within journal limits. Please note that page charges are made on all papers longer than 20 pages. If you cannot pay these charges you must reduce your paper to 20 pages before submitting your revision. Your paper has been ESTIMATED to be 28 pages. We cannot proceed with typesetting your paper without your agreement to meet page charges in full should the paper exceed 20 pages when typeset. If you have any questions, please do get in touch.

To upload a resubmitted manuscript, log into <http://mc.manuscriptcentral.com/prsa> and enter your Author Centre, where you will find your manuscript title listed under "Manuscripts with Decisions." Under "Actions," click on "Create a Resubmission." Please be sure to indicate that it is a resubmission, and ensure you enter this ID - RSPA-2020-0328 - as the previous submission number.

Yours sincerely
 Raminder Shergill
 proceedingsa@royalsociety.org

Reviewer(s)' Comments to Author:

Referee: 1

Comments to the Author(s)

This paper introduces a probabilistic approach to assess the tsunami currents hazards in the nearshore based on a statistical method, called an emulator. Such a 'new' type of tool is expected to provide an efficient scheme for analyzing tsunami hazards and to help break through barriers that other PTHA based on numerical modeling may have. Meanwhile, there are major issues to be responded to before this work is published, as listed below.

1. Throughout the entire manuscript, figures weren't presented in an appropriate way, too many windows and information condensed in a single figure slot. This led to lose the focus of each figure. For example, Fig. 5 & 7 can be re-arranged. Figure 6 is not so significant in the main body of the manuscript and can be relocated in either appendix or supporting material if applicable. Fig. 8 and others can be lessened by removing some sub-plots or by merging some of them. So I strongly encourage the author to review their figures and evaluate them to determine which are crucial to be shown in and also separate them into two or more. This will increase the visibility of the included figures and aid the reader to follow them easily.
2. The author can put more stress on Section 2. (e), (f), (g) since they contain a huge part of novelty (i.e., emulation scheme) of the present study. Such an effort will distinguish what is typical and what is new in this research. I recommend the author separate them from Section 2 in some way.
3. Abstract needs to be re-drafted to concisely address the significance of this study. Currently stated are the modeling process, which is not able to deliver and efficiently highlight the main contribution of this study to the tsunami research. Thus, the rewritten abstract should state the main steam of the study, clearly declaring the contribution of the main findings that had been achieved.
4. Introduction is required to be reshaped. For example, a detailed description of what happened at the Salalah port during the 2004 Sumatra-Andaman tsunami can be omitted. Rather than including that, the authors can endeavor to top up more reviews on probabilistic methods since the core part of this study is how to access the tsunami hazards on a statistical basis. Therefore, more literatures stating the probabilistic assessment of tsunami hazards need to be thoroughly and comprehensively reviewed and added (e.g., Zhang and Niu (2020)).
5. It is mentioned that "the strong currents continued for hours after the waves with maximum amplitude had arrived (nearly 9h in Salalah.)" (Page 2 line 31). This long-lasting currents simulation can be achieved only when the higher-order scheme guaranteeing the least numerical diffusion is applied. Therefore, numerical dissipation embedded in the applied model, VOLNA-OP2, which may corrupt the modeling results needs to be justified along with a wide range of reviews on other tsunami models in this regard. Moreover, the uncertainty originating from using a specific type of model should be examined. Lynett et al. (2017) provided the comprehensive simulation results from various numerical models focusing on the tsunami-current generation and inter-compared them. It suggested that the huge uncertainties may result

from using different types of numerical models, particularly when a high-definition grid less than 10 m is applied.

6. What is 'rupture origin' Did the authors mean an epicenter by mentioning it? If so, an epicenter must be a better expression. Also, the terms, 'fault' and 'rupture' were being used confusingly in several parts, which is not acceptable in the scientific journal. Please make a clear definition of each ahead of its use and use it appropriately when required.

7. The description of 300 earthquake scenarios is not enough for readers to figure them out. The authors need to detail how the 300 cases had been constructed, through a sort of the table showing the entire combination of themselves. The algorithm to generate 300 scenarios based on Latin Hypercube Design should also have physical sense. Moreover, developing a million predictions from the 300 cases is not well understood despite given explanation and figures, and thus conclusions drawn from those results are not so convincing.

8. The sediment amplification and the resulting bathymetric change introduced by Eq. (2.11) seem to lost lots of physical standard. For example, there must be another set of equations for bathymetric changes coupled to Eq. (2.7) and (2.8) (e.g., Li et al. (2012), Jaffe et al. (2016)). Please address how the authors handled this issue.

9. When the bathymetries from several sources are merged, special care needs to be paid because they may have different datum. Please explain how the author handled this issue.

10. How do the authors define 'high-definition' of resolution in bathy, mesh, and coastline? It seems it refers O(10m), but there have been similar achievements up to those resolutions so far and it is not a quite challenging issue.

11. Figure 2: It is not easy to catch the main flow of Figure 2. Further simplification and conceptualization are required for the elegant representation. Also, the section number included in the figure didn't match with one in the main body.

13. Please cite Yuan et al. (2020) where applicable, since it provided good accomplishment of using GPU in tsunami modeling.

Jaffe B, Goto K, Sugawara D, Gelfenbaum G, La Selle S. 2016. Uncertainty in tsunami sediment transport modeling. *Journal of Disaster Research* 11(4): 647-661.

Li L, Qiu Q, Huang Z. 2012. Numerical modeling of the morphological change in Lhok Nga, west Banda Aceh, during the 2004 Indian Ocean tsunami: understanding tsunami deposits using a forward modeling method. *Natural Hazards* 64: 1549-1574.

Lynett, P. J., Gately, K., Wilson, R., Montoya, L., Arcas, D., Aytore, B., ... & David, C. G. (2017). Inter-model analysis of tsunami-induced coastal currents. *Ocean Modelling*, 114, 14-32.

Yuan, Y., Shi, F., Kirby, J. T., & Yu, F. (2020). FUNWAVE-GPU: Multiple-GPU Acceleration of a Boussinesq-Type Wave Model. *Journal of Advances in Modeling Earth Systems*, 12(5), e2019MS001957.

Zhang, X., & Niu, X. (2020). Probabilistic tsunami hazard assessment and its application to southeast coast of Hainan Island from Manila Trench. *Coastal Engineering*, 155, 103596.

Referee: 2

Comments to the Author(s)

Review of the manuscript: "Probabilistic Quantification of Tsunami Current Hazard using Statistical Emulation" by D. Gopinthan et al.

The manuscript presents a method to evaluate the probabilistic hazard of tsunami currents in specific target sites (e.g., harbours), using statistical emulators to reduce the computational resources. Moreover, the work is focused on tsunami currents which present some differences in terms of hazard with respect to the more common use of tsunami wave heights as hazard intensity measure.

The topic is of interest for the tsunami community and I agree with authors that both the method (statistical emulator) and the kind of intensity measure (tsunami currents) are still not much explored in tsunami science.

However, I have some comments that, in my opinion, should be addressed before the publications.

First of all, I have a very general comment.

My main concern is the possible misleading of the actual goal of the manuscript: in its present form, the work is illustrative of a general workflow that can be used to assess probabilistic tsunami hazard. Each step is technically detailed in its implementation and this is a really worthy effort from the authors. But, on the other hand, I would stress that they should be careful to avoid that a reader could be confused that this could be an example of hazard assessment. To be clearer: immediately in the abstract it is stated that the work "[...] map probabilistic representations of maximum tsunami velocities and heights for the full range of possible future tsunamis at around 200 locations around Karachi port, Makran Subduction Zone (MSZ)". In my opinion, this is not true. A rigorous source analysis from the seismic and tectonic point of view is missing, so I disagree with authors that the "full range" of possibilities is here considered. For this reason, I suggest that in some part of the text (likely in the title itself? "A method to quantify..." or something like this?) authors should try to stress more that this work is purely illustrative of a method.

My thoughts are partially confirmed by the authors themselves: indeed, in the discussion, they are more prudent, with a list of missing sources of uncertainty that should be considered before performing "a full probabilistic assessment" (page 21, lines 51 and following): scaling relations, G-R maximum moment magnitudes, bathymetry. So they are in part aware of this.

Moreover, I think that also the seismic source variability is only partially explored and poorly described; in my opinion, this is an essential component of such hazard analyses and should at least be presented. Indeed, even though the case study is illustrative, the MSZ source zone and the Karachi Port are real, so I think that it is important to introduce the geophysical context and better address some input choices.

I'm not asking to explicitly explore the full source variability: but I think that authors should, at least, add an exhaustive section to introduce the tectonic setting of the area and better explain that they are considering only a part of the potential sources for illustrative purposes.

Here, for example, the authors are excluding the western part of the MSZ subduction, which is already a quite important choice (see my comment #1). Then, what about other potential tsunamigenic seismic sources? Could splay faults, outer-rise earthquakes and crustal earthquakes generate tsunamis with impact for the Karachi port? Could these sources be easily incorporated in the present procedure or they represent a challenge? Please enrich the text with this part.

Finally, some subjective choices are just presented as they are, without explanations: this could lead a reader to think that these could be common choices, but they aren't (see my comment #2 about the range of magnitudes).

Other comments:

1. Why excluding the western part of MSZ? It seems to me that the illustrative example could be more solid if applied to the whole subduction instead to a part of it. The considered G-R distribution refers to the whole subduction or just to the eastern part? Could you please address this? I wasn't able to find Table S1 (mentioned at page 17, lines 34-35) so I could not check how the authors selected the b-value. However, if this extension to the western MSZ does not present technical issues, why do not model it? The results would be representative of the whole subduction and could be more interesting to comment.
2. Why the minimum considered magnitude is 7.5? for saving numbers because of illustrative purposes or there is any geophysical reason to neglect magnitudes < 7.5 ? Small tsunamis couldn't generate currents as well? Please address this point.
3. I found that the state of the art is not complete. For example, other works describing general methods to propagate the uncertainty and/or to reduce the number of tsunami simulations required to perform probabilistic tsunami hazards have been published (Lorito et al., 2015, Volpe et al., 2019 and very recently Kotani et al., 2020 could be of interest). I think that, even though it was on landslide-generated tsunami, the work on statistical emulators by Sarri et al. in 2012 represents an existing reference of a similar approach and merit be mentioned.
4. Authors could re-think the manuscript's structure, for example by moving to Appendix some of the most technical parts, in order to lighten the reading of the main text, which is now very dense of figures and subfigures that could distract the reader. One explicit example is the following: Appendices A, B and C have a corresponding section with the same title also in the main text (Slip profile generation, Merging of Bathymetries and Localised Non-Uniform Unstructured Mesh); I think that this is redundant: I suggest to keep the Appendices only and also move there the corresponding figures.
5. This is just a suggestion for a future implementation: a further module that could be incorporated in the procedure is the effect of the water layer between the seafloor displacement and the corresponding water surface perturbation (see Kajiura, 1963).

Some minor comments:

- I can't find any table mentioned in the text. Maybe there was an issue during files' submission?
- Introduction, page 2, line 24: A $M_w=7.7$ earthquake is not small. I think authors can remove the adjective "small" which is not necessary (and not understandable) in this sentence.
- Introduction, page 2, line 26: I cannot find Table 1 and reference to Figure 3 seems not correct, please check.
- Introduction, page 2, line 28: I suggest to use labels when referring to subplots (ad example Figure 1b) to help the reader.
- Why, in figures 4, 5 and 9, the 3000 m depth contour line is explicitly shown? It seems to me a little confusing with the coastline. Is it necessary in these plots? If not, you could remove it and keep only the colormap in figure 7a for water depth.

References:

Kajiura K., 1963. The leading wave of a tsunami, Bull. Earthq. Res. Inst. Univ. Tokyo, 41, 535-571

Kotani T., K. Tozato, S. Takase, S. Moriguchi, K. Terada, Y. Fukutani, Y. Otake, K. Nojima, M. Sakuraba, Y. Choe, Probabilistic tsunami hazard assessment with simulation-based response surfaces, *Coastal Engineering*, 160, 2020, doi:10.1016/j.coastaleng.2020.103719

Lorito S., Selva, J., Basili, R., Romano, F., Tiberti, M. M., & Piatanesi, A. (2015). Probabilistic hazard for seismically induced tsunamis: Accuracy and feasibility of inundation maps. *Geophysical Journal International*, 200(1), 574–588. <https://doi.org/10.1093/gji/ggu408>

Sarri, A., Guillas, S., and Dias, F.: Statistical emulation of a tsunami model for sensitivity analysis and uncertainty quantification, *Nat. Hazards Earth Syst. Sci.*, 12, 2003–2018, <https://doi.org/10.5194/nhess-12-2003-2012>, 2012

Volpe, M., Lorito, S., Selva, J., Tonini, R., Romano, F., and Brizuela, B.: From regional to local SPTHA: efficient computation of probabilistic tsunami inundation maps addressing near-field sources, *Nat. Hazards Earth Syst. Sci.*, 19, 455–469, <https://doi.org/10.5194/nhess-19-455-2019>, 2019.

Author's Response to Decision Letter for (RSPA-2020-0328.R0)

See Appendices A & B.

RSPA-2021-0180.R0

Review form: Referee 1

Is the manuscript an original and important contribution to its field?

Acceptable

Is the paper of sufficient general interest?

Acceptable

Is the overall quality of the paper suitable?

Acceptable

Do you have any ethical concerns with this paper?

No

Recommendation?

Accept as is

Comments to the Author(s)

The revised manuscript had gone through huge modifications to address the issues that I gave in the previous review. I'm satisfied with what the authors had done, and recommend the publication of this manuscript as it is.

Review form: Referee 2

Is the manuscript an original and important contribution to its field?

Good

Is the paper of sufficient general interest?

Good

Is the overall quality of the paper suitable?

Acceptable

Do you have any ethical concerns with this paper?

No

Recommendation?

Major revision is needed (please make suggestions in comments)

Comments to the Author(s)

This is my second review of the manuscript "Probabilistic Quantification of Tsunami Current Hazard using Statistical Emulation" by Gopinathan et al.

The authors addressed most of the comments raised after the first round review and made a notable work to revise the manuscript, but I think that some work is still required before recommending it for publication. However, the authors should not feel discouraged by the "Major revision" status, since most of my comments are, this second time, about the structure of the manuscript only.

In general, I found quite atypical to mention figures in the text without following the order in which they were displayed: for example, figure 7 is called in the text before figures 3-to-6 and figure 11 is called before figures 6-to-10. In my opinion, this situation reflects a still partial confusing way to present the work. This mainly happens in Introduction and in Section 2; the latter, in particular, does not properly describe the workflow scheme.

Therefore, I suggest to consider a further re-arrangement of the manuscript in order to make the reading more linear.

Here below, I will try to better clarify my thoughts with a list of more specific comments (in the following, page numbers refer to the numbering in the black boxes on top of the pdf document):

1. In my opinion, it could be of help rearranging the Introduction by introducing all general arguments before introducing the (specific) case study of Makran. For example: the sentence "The Makran Subduction Zone ... to be Mw 8.8-9.0 [13]." (page 2, lines 49-52) could be moved below, since the "pressing need for a comprehensive quantification of tsunami hazard" is a general issue. Also the new paragraph about probabilistic methods (page 3, lines 15-27) is a general argument and could be placed before introducing the Makran case study. Starting from a wider perspective and progressively narrowing the focus is often a good strategy to introduce the presented research.
2. I would separate Figure 1 in, at least, two figures: I think that the proposed workflow should have a dedicated figure and should be introduced, in my opinion, in section 2.
3. About the workflow scheme: in my understanding, the design's box should be independent on sampling (the light blue box). Instead, the LHD sampling (300 events) should be in input to the emulation construction and the MOCD sampling (1 million events) should be in input to the

emulator prediction. Please address if I am misunderstanding the workflow.

4. Sections 2 (and in part also section 3, see comment #5) should be better organized: my best option would be to keep separated the general description of the workflow (description of the single modules' functionalities together with the technical details in the corresponding appendices) from the specific set up used for Makran. But if this decoupling is not preferred by authors, I suggest to organize it in the most concise (but general) way and, then, follow it through the section and subsections, maintaining (or expanding if needed) the appendices for the more technical parts and keeping the main text fluent.

Example: the effective displacement U is introduced at page 5, referring at Figure 7, while a few lines above the effective slip S is described using the Figure 2. This forces the reader to jump five figures ahead and back. A possible solution could be to incorporate the corresponding U snapshot in Figure 2, so the whole construction of the initial profile would be shown in one figure (and figure 7 could be removed). And this could become a "Tsunami source" subsection (preceding the "Tsunami propagation" one) in which introducing all the sub-steps used to create the initial profile (finite fault, slip profile, sediments, deformation, uplift).

I tried an example of section 2 as closest as possible to what already proposed by authors, but in my view:

2 data and methods

2.1 makran subduction zone

2.2 tsunami source

- 2.2a finite fault

- 2.2b slip profile

- 2.2c sediments' amplification

- 2.2d vertical displacement

2.3 tsunami propagation

The "makran subduction zone" section could even stay in a separate section as well (becoming section 2 and scaling all the rest consequently): however, the main point for me is that the "makran subduction zone" section should contain the general description of the case study only, moving the construction of the finite fault (which is one of the steps in preparing the tsunami source) in a dedicated subsection.

5. Independently on how would be shaped the final workflow scheme, I recommend to move the G-R definition in the "makran subduction zone" section, since the G-R is something which characterize the area of interests (seismicity and rates), similarly to the tectonic setting. Then, G-R can be simply recalled in section 3.1 and 3.3 only to describe how the magnitude is sampled in the two phases of emulation (construction and prediction).

6. All the figures moved (or that will be moved) in the appendices should be numbered differently with respect the figures in the main text (for example, A_x , A_y , B_x , etc). This helps in discerning between what can be deepened in a second moment (for example, at the beginning of section 2, when figure 15 is called).

Other minor comments:

Page 3, line 15: I suggest to use "Probabilistic scenario-based ..." as a few lines below, as counterpart of "deterministic scenario-based"

Page 7, line 15: I suggest to specify, at the end of this sentence, that authors are referring to the Makran area: "... is considerably higher than in existing studies related to MSZ [41-43]".

Page 17, lines 37-38: As the previous comment, I suggest to specify that this sentence refers to the Makran area: "For MSZ, the major influence of maximum magnitude was illustrated ..."

If it could be useful to reduce the number of figures in the main text, I would suggest to remove panels b and d from Figures 4 and 5 and join the remaining panels 4a, 4c, 5a and 5c.

Review form: Referee 3

Is the manuscript an original and important contribution to its field?

Good

Is the paper of sufficient general interest?

Good

Is the overall quality of the paper suitable?

Good

Can the paper be shortened without overall detriment to the main message?

Yes

Do you think some of the material would be more appropriate as an electronic appendix?

No

Do you have any ethical concerns with this paper?

No

Recommendation?

Accept with minor revision (please list in comments)

Comments to the Author(s)

See attached files

Decision letter (RSPA-2021-0180.R0)

22-Apr-2021

Dear Dr Gopinathan,

On behalf of the Editor, I am pleased to inform you that your Manuscript RSPA-2021-0180 entitled "Probabilistic Quantification of Tsunami Current Hazard using Statistical Emulation" has been accepted for publication subject to minor revisions in Proceedings A. Please find the referees' comments below.

The reviewer(s) have recommended publication, but also suggest some minor revisions to your manuscript. Therefore, I invite you to respond to the reviewer(s)' comments and revise your manuscript. Please note that we have a strict upper limit of 28 pages for each paper. Please endeavour to incorporate any revisions while keeping the paper within journal limits. Please note that page charges are made on all papers longer than 20 pages. If you cannot pay these charges you must reduce your paper to 20 pages before submitting your revision. Your paper has been ESTIMATED to be 28 pages. We cannot proceed with typesetting your paper without your agreement to meet page charges in full should the paper exceed 20 pages when typeset. If you have any questions, please do get in touch.

It is a condition of publication that you submit the revised version of your manuscript within 7 days. If you do not think you will be able to meet this date please let me know in advance of the due date.

To revise your manuscript, log into <https://mc.manuscriptcentral.com/prsa> and enter your Author Centre, where you will find your manuscript title listed under "Manuscripts with Decisions." Under "Actions," click on "Create a Revision." Your manuscript number has been appended to denote a revision.

You will be unable to make your revisions on the originally submitted version of the manuscript. Instead, revise your manuscript and upload a new version through your Author Centre.

When submitting your revised manuscript, you will be able to respond to the comments made by the referee(s) and upload a file "Response to Referees" in Step 1: "View and Respond to Decision Letter". You can use this to document any changes you make to the original manuscript. In order to expedite the processing of the revised manuscript, please be as specific as possible in your response to the referee(s).

IMPORTANT: Your original files are available to you when you upload your revised manuscript. Please delete any redundant files before completing the submission process.

When uploading your revised files, please make sure that you include the following as we cannot proceed without these:

- 1) A text file of the manuscript (doc, txt, rtf or tex), including the references, tables (including captions) and figure captions. Please remove any tracked changes from the text before submission. PDF files are not an accepted format for the "Main Document".
- 2) A separate electronic file of each figure (tif, eps or print-quality pdf preferred). The format should be produced directly from original creation package, or original software format.
- 3) Electronic Supplementary Material (ESM): all supplementary materials accompanying an accepted article will be treated as in their final form. Note that the Royal Society will not edit or typeset supplementary material and it will be hosted as provided. Please ensure that the supplementary material includes the paper details where possible (authors, article title, journal name). Supplementary files will be published alongside the paper on the journal website and posted on the online figshare repository (<https://figshare.com>). The heading and legend provided for each supplementary file during the submission process will be used to create the figshare page, so please ensure these are accurate and informative so that your files can be found in searches. Files on figshare will be made available approximately one week before the accompanying article so that the supplementary material can be attributed a unique DOI. Alternatively you may upload a zip folder containing all source files for your manuscript as described above with a PDF as your "Main Document". This should be the full paper as it appears when compiled from the individual files supplied in the zip folder.

Article Funder

Please ensure you fill in the Article Funder question on page 2 to ensure the correct data is collected for FundRef (<http://www.crossref.org/fundref/>).

Media summary

Please ensure you include a short non-technical summary (up to 100 words) of the key findings/importance of your paper. This will be used for to promote your work and marketing purposes (e.g. press releases). The summary should be prepared using the following guidelines:

*Write simple English: this is intended for the general public. Please explain any essential technical terms in a short and simple manner.

*Describe (a) the study (b) its key findings and (c) its implications.

*State why this work is newsworthy, be concise and do not overstate (true 'breakthroughs' are a rarity).

*Ensure that you include valid contact details for the lead author (institutional address, email address, telephone number).

Cover images

We welcome submissions of images for possible use on the cover of Proceedings A. Images should be square in dimension and please ensure that you obtain all relevant copyright permissions before submitting the image to us. If you would like to submit an image for consideration please send your image to proceedingsa@royalsociety.org

Open Access

You are invited to opt for open access, our author pays publishing model. Payment of open access fees will enable your article to be made freely available via the Royal Society website as soon as it is ready for publication. For more information about open access please visit <https://royalsociety.org/journals/authors/open-access/>. The open access fee for this journal is £1700/\$2380/€2040 per article. VAT will be charged where applicable. Please note that if the corresponding author is at an institution that is part of a Read and Publishing deal you are required to select this option. See <https://royalsociety.org/journals/librarians/purchasing/read-and-publish/read-publish-agreements/> for further details.

Once again, thank you for submitting your manuscript to Proceedings A and I look forward to receiving your revision. If you have any questions at all, please do not hesitate to get in touch.

Best wishes
Raminder Shergill
proceedingsa@royalsociety.org
Proceedings A

Reviewer(s)' Comments to Author:

Referee: 3

Comments to the Author(s)

See attached files

Referee: 1

Comments to the Author(s)

The revised manuscript had gone through huge modifications to address the issues that I gave in the previous review. I'm satisfied with what the authors had done, and recommend the publication of this manuscript as it is.

Referee: 2

Comments to the Author(s)

This is my second review of the manuscript "Probabilistic Quantification of Tsunami Current Hazard using Statistical Emulation" by Gopinathan et al.

The authors addressed most of the comments raised after the first round review and made a notable work to revise the manuscript, but I think that some work is still required before recommending it for publication. However, the authors should not feel discouraged by the "Major revision" status, since most of my comments are, this second time, about the structure of the manuscript only.

In general, I found quite atypical to mention figures in the text without following the order in which they were displayed: for example, figure 7 is called in the text before figures 3-to-6 and figure 11 is called before figures 6-to-10. In my opinion, this situation reflects a still partial confusing way to present the work. This mainly happens in Introduction and in Section 2; the latter, in particular, does not properly describe the workflow scheme.

Therefore, I suggest to consider a further re-arrangement of the manuscript in order to make the reading more linear.

Here below, I will try to better clarify my thoughts with a list of more specific comments (in the following, page numbers refer to the numbering in the black boxes on top of the pdf document):

1. In my opinion, it could be of help rearranging the Introduction by introducing all general arguments before introducing the (specific) case study of Makran. For example: the sentence "The Makran Subduction Zone ... to be Mw 8.8-9.0 [13]." (page 2, lines 49-52) could be moved below, since the "pressing need for a comprehensive quantification of tsunami hazard" is a general issue. Also the new paragraph about probabilistic methods (page 3, lines 15-27) is a general argument and could be placed before introducing the Makran case study. Starting from a wider perspective and progressively narrowing the focus is often a good strategy to introduce the presented research.
2. I would separate Figure 1 in, at least, two figures: I think that the proposed workflow should have a dedicated figure and should be introduced, in my opinion, in section 2.
3. About the workflow scheme: in my understanding, the design's box should be independent on sampling (the light blue box). Instead, the LHD sampling (300 events) should be in input to the emulation construction and the MOCD sampling (1 million events) should be in input to the emulator prediction. Please address if I am misunderstanding the workflow.
4. Sections 2 (and in part also section 3, see comment #5) should be better organized: my best option would be to keep separated the general description of the workflow (description of the single modules' functionalities together with the technical details in the corresponding appendices) from the specific set up used for Makran. But if this decoupling is not preferred by authors, I suggest to organize it in the most concise (but general) way and, then, follow it through the section and subsections, maintaining (or expanding if needed) the appendices for the more technical parts and keeping the main text fluent.

Example: the effective displacement U is introduced at page 5, referring at Figure 7, while a few lines above the effective slip S is described using the Figure 2. This forces the reader to jump five figures ahead and back. A possible solution could be to incorporate the corresponding U snapshot in Figure 2, so the whole construction of the initial profile would be shown in one figure (and figure 7 could be removed). And this could become a "Tsunami source" subsection (preceding the "Tsunami propagation" one) in which introducing all the sub-steps used to create the initial profile (finite fault, slip profile, sediments, deformation, uplift).

I tried an example of section 2 as closest as possible to what already proposed by authors, but in my view:

2 data and methods

2.1 makran subduction zone

2.2 tsunami source

- 2.2a finite fault

- 2.2b slip profile

- 2.2c sediments' amplification

- 2.2d vertical displacement

2.3 tsunami propagation

The "makran subduction zone" section could even stay in a separate section as well (becoming section 2 and scaling all the rest consequently): however, the main point for me is that the "makran subduction zone" section should contain the general description of the case study only, moving the construction of the finite fault (which is one of the steps in preparing the tsunami source) in a dedicated subsection.

5. Independently on how would be shaped the final workflow scheme, I recommend to move the G-R definition in the "makran subduction zone" section, since the G-R is something which characterize the area of interests (seismicity and rates), similarly to the tectonic setting. Then, G-R can be simply recalled in section 3.1 and 3.3 only to describe how the magnitude is sampled in the two phases of emulation (construction and prediction).

6. All the figures moved (or that will be moved) in the appendices should be numbered differently with respect the figures in the main text (for example, Ax, Ay, Bx, etc). This helps in discerning between what can be deepened in a second moment (for example, at the beginning of section 2, when figure 15 is called).

Other minor comments:

Page 3, line 15: I suggest to use "Probabilistic scenario-based ... " as a few lines below, as counterpart of "deterministic scenario-based"

Page 7, line 15: I suggest to specify, at the end of this sentence, that authors are referring to the Makran area: " ... is considerably higher than in existing studies related to MSZ [41-43]".

Page 17, lines 37-38: As the previous comment, I suggest to specify that this sentence refers to the Makran area: "For MSZ, the major influence of maximum magnitude was illustrated ..."

If it could be useful to reduce the number of figures in the main text, I would suggest to remove panels b and d from Figures 4 and 5 and join the remaining panels 4a, 4c, 5a and 5c.

Decision letter (RSPA-2021-0180.R1)

10-May-2021

Dear Dr Gopinathan

I am pleased to inform you that your manuscript entitled "Probabilistic Quantification of Tsunami Current Hazard using Statistical Emulation" has been accepted in its final form for publication in Proceedings A.

Our Production Office will be in contact with you in due course. You can expect to receive a proof of your article soon. Please contact the office to let us know if you are likely to be away from e-mail in the near future. If you do not notify us and comments are not received within 5 days of sending the proof, we may publish the paper as it stands.

As a reminder, you have provided the following 'Data accessibility statement' (if applicable). Please remember to make any data sets live prior to publication, and update any links as needed when you receive a proof to check. It is good practice to also add data sets to your reference list.
Statement (if applicable): The data and codes used have been cited and/or linked in footnotes on first mention.

Open access

You are invited to opt for open access, our author pays publishing model. Payment of open access fees will enable your article to be made freely available via the Royal Society website as soon as it is ready for publication. For more information about open access please visit <https://royalsociety.org/journals/authors/which-journal/open-access/>. The open access fee for this journal is £1700/\$2380/€2040 per article. VAT will be charged where applicable.

Note that if you have opted for open access then payment will be required before the article is published – payment instructions will follow shortly.

If you wish to opt for open access then please inform the editorial office (proceedingsa@royalsociety.org) as soon as possible.

Your article has been estimated as being 28 pages long. Our Production Office will inform you of the exact length at the proof stage.

Proceedings A levies charges for articles which exceed 20 printed pages. (based upon approximately 540 words or 2 figures per page). Articles exceeding this limit will incur page charges of £150 per page or part page, plus VAT (where applicable).

Under the terms of our licence to publish you may post the author generated postprint (ie. your accepted version not the final typeset version) of your manuscript at any time and this can be made freely available. Postprints can be deposited on a personal or institutional website, or a recognised server/repository. Please note however, that the reporting of postprints is subject to a media embargo, and that the status the manuscript should be made clear. Upon publication of the definitive version on the publisher's site, full details and a link should be added.

You can cite the article in advance of publication using its DOI. The DOI will take the form: 10.1098/rspa.XXXX.YYYY, where XXXX and YYYY are the last 8 digits of your manuscript number (eg. if your manuscript number is RSPA-2017-1234 the DOI would be 10.1098/rspa.2017.1234).

For tips on promoting your accepted paper see our blog post: <https://royalsociety.org/blog/2020/07/promoting-your-latest-paper-and-tracking-your-results/>

On behalf of the Editor of Proceedings A, we look forward to your continued contributions to the Journal.

Sincerely,
Raminder Shergill
proceedingsa@royalsociety.org

Appendix A

RESPONSE TO REFEREE #1

This paper introduces a probabilistic approach to assess the tsunami currents hazards in the nearshore based on a statistical method, called an emulator. Such a ‘new’ type of tool is expected to provide an efficient scheme for analyzing tsunami hazards and to help break through barriers that other PTHA based on numerical modeling may have. Meanwhile, there are major issues to be responded to before this work is published, as listed below.

Reply – The authors thank the referee for the comments which have not only improved the content of the manuscript, but also refined its layout.

A point-wise reply is included below. Significant modifications to passages in the revised manuscript now appear in red. An exception is when whole sections/sub-sections have undergone modification, in which case the section/sub-section title appears in red.

1. Throughout the entire manuscript, figures weren’t presented in an appropriate way, too many windows and information condensed in a single figure slot. This led to lose the focus of each figure. For example, Fig. 5 & 7 can be re-arranged. Figure 6 is not so significant in the main body of the manuscript and can be relocated in either appendix or supporting material if applicable. Fig. 8 and others can be lessened by removing some sub-plots or by merging some of them. So I strongly encourage the author to review their figures and evaluate them to determine which are crucial to be shown in and also separate them into two or more. This will increase the visibility of the included figures and aid the reader to follow them easily.

Reply – We thank the referee for this suggestion. It has immensely helped in streamlining the layout of the content. All the figures have been reworked to effect this change. Old Figures. 3, 4, 5, 7, 8, 9, 10 and 14 have been split/lessened, sub-labels reduced or merged. Old Figures. 3b, 4, 6, and 7 have been moved to the appendix for ease of reading of the main text.

2. The author can put more stress on Section 2. (e), (f), (g) since they contain a huge part of novelty (i.e., emulation scheme) of the present study. Such an effort will distinguish what is typical and what is new in this research. I recommend the author separate them from Section 2 in some way.

Reply – The referee’s recommendation was helpful in streamlining the sections. We agree with the referee that the emulation comprises the major component of the novelty in this work. Thus, as per the recommendation, we have moved the sub-sections related to emulator construction, diagnostics and prediction to the newly created Section 3 (Statistical Emulation). Further, the sub-sections on emulator construction (Section 3a) and diagnostics

(Section 3b) have been expanded for more clarity with descriptions of the emulator code (MOGP) and leave-one-out (L-O-O) procedure respectively. Additionally, we have also refined Section 2 (Models, Data and Methods) by relocating the descriptions of the complex, but less innovative work of slip profile construction, bathymetry merging and meshing algorithm to the Appendices A, B and C respectively.

3. Abstract needs to be re-drafted to concisely address the significance of this study. Currently stated are the modeling process, which is not able to deliver and efficiently highlight the main contribution of this study to the tsunami research. Thus, the rewritten abstract should state the main steam of the study, clearly declaring the contribution of the main findings that had been achieved.

Reply – We comply with the referee’s comment, the abstract has been re-drafted to shift the focus from the numerical modelling to the contributions arising from the use of emulation.

4. Introduction is required to be reshaped. For example, a detailed description of what happened at the Salalah port during the 2004 Sumatra-Andaman tsunami can be omitted. Rather than including that, the authors can endeavor to top up more reviews on probabilistic methods since the core part of this study is how to access the tsunami hazards on a statistical basis. Therefore, more literatures stating the probabilistic assessment of tsunami hazards need to be thoroughly and comprehensively reviewed and added (e.g., Zhang and Niu (2020)).

Reply – In accordance with the referee’s suggestions, the introduction has been considerably revised and references added. We have removed the detailed description of what happened at the Salalah port during the 2004 Sumatra-Andaman tsunami, and added a review of probabilistic methods in the third paragraph of Section 1. The sub-figures related to Salalah port have been removed from Figure. 1. The text now also leans more towards the emulation.

5. It is mentioned that “the strong currents continued for hours after the waves with maximum amplitude had arrived (nearly 9h in Salalah.)” (Page 2 line 31). This long-lasting currents simulation can be achieved only when the higher-order scheme guaranteeing the least numerical diffusion is applied. Therefore, numerical dissipation embedded in the applied model, VOLNA-OP2, which may corrupt the modeling results needs to be justified along with a wide range of reviews on other tsunami models in this regard. Moreover, the uncertainty originating from using a specific type of model should

be examined. Lynett et al. (2017) provided the comprehensive simulation results from various numerical models focusing on the tsunami-current generation and inter-compared them. It suggested that the huge uncertainties may result from using different types of numerical models, particularly when a high-definition grid less than 10 m is applied.

Reply – We concur with the referee and have added the below points in Section 2(c). Numerical dissipation is a major factor affecting the fidelity of the simulation of long-lasting currents. Thus, while the inter-model comparison in Lynett et al. [1] is an important benchmark that highlights the pitfalls in attempting high-resolution current simulations, we limit our numerical studies using VOLNA-OP2 in line with the scope of the work. In this work, our stress is on the framework of emulation which can be utilized in conjunction with any other tsunami solver. Much can be done to minimize numerical dissipation in VOLNA-OP2. Towards this, we cite Giles et al. [2], wherein the numerical errors in VOLNA-OP2 are analysed by decomposing them into dispersion and dissipation components.

6. What is ‘rupture origin’ Did the authors mean an epicenter by mentioning it? If so, an epicenter must be a better expression. Also, the terms, ‘fault’ and ‘rupture’ were being used confusingly in several parts, which is not acceptable in the scientific journal. Please make a clear definition of each ahead of its use and use it appropriately when required.

Reply – We thank the referee for pointing this error. Occurrences of *rupture origin* have been replaced with *epicenter*, as per referee’s advice. Further, the term *fault* has been consistently used throughout.

7. The description of 300 earthquake scenarios is not enough for readers to figure them out. The authors need to detail how the 300 cases had been constructed, through a sort of the table showing the entire combination of themselves. The algorithm to generate 300 scenarios based on Latin Hypercube Design should also have physical sense. Moreover, developing a million predictions from the 300 cases is not well understood despite given explanation and figures, and thus conclusions drawn from those results are not so convincing.

Reply –

300 training cases - We acknowledge this important point made by the referee, and have complied by creating two supplementary animations. They show the exact finite-fault configuration, slip profile, and seabed deformation for the 300 cases (with and without the influence of sediments). They also portray how the Latin hypercube sampling sweeps through the input parameter space, and the manner in which the dimensions of the 300 sources are

scaled with respect to the scaling relation. Additionally, The parameters used to create the 300 sources are shown in Figure 6, illustrated with two scenarios in Figure 7. The new Appendix A also describes how the input parameters (M_w, X_o, Y_o) generate the slip profiles. Further, the referee’s concern about the lack of information detailing the 300 cases is addressed by confirming that the 300 finite-fault earthquake sources respect the plate-boundary geometry in the latest Slab2 model (also mentioned in the manuscript).

Latin Hypercube Design (LHD) - We thank the referee for this insightful comment and address this by including the following explanation in the second paragraph of Section 3(a). The specific purpose of the design stage is to capture the functional relationship between the input parameters (M_w, X_o, Y_o) and output quantities (η_{max}, v_{max}) at a location. The LHD generates a set of points that are nearly uniformly spread to cover the input parameter space. Specifically, it maximises the minimum distance between points in the set, a feature that explores the functional relationship better than a random scatter. In a physical sense, this spread of points endeavours to capture the information inherent in the input-output relationship as much as possible.

1 million predictions - We address the referee’s concern by adding the following description in the second paragraph of Section 3(a). The Gaussian Process (GP) emulator interpolates across the input-output points in the training set. In other words, the constructed emulator works as an approximation, and can be used to generate predictions (or, evaluated) at any point in the space of input parameters. The predictions will be exact at the training points, but uncertain elsewhere. This uncertainty is modeled by a normal distribution whose mean and standard deviation are calculated using the Kriging formula.

8. The sediment amplification and the resulting bathymetric change introduced by Eq. (2.11) seem to lost lots of physical standard. For example, there must be another set of equations for bathymetric changes coupled to Eq. (2.7) and (2.8) (e.g., Li et al. (2012), Jaffe et al. (2016)). Please address how the authors handled this issue.

Reply – As the referee indicates, indeed, the amplification due to presence of sediments needs better modelling. We assent by including the following clarification in the first paragraph of Section 2(b). Incorporation of the effect of sediments influence tsunami modelling mainly in two ways. First, the interplay of sediment transport and tsunami flow gives rise to enhanced coupled morph- and hydro-dynamics [3, 4]. Second, the Okada deformation model [5], with the assumptions of an elastic, homogeneous, isotropic medium in a semi-infinite domain, can be improved by sediment models that exhibit

non-linear, non-homogeneous, and an-isotropic behaviour. Considerable amplification (up to 60% locally) of crustal deformation due to the presence of layers of sediments on the seafloor can occur [6]. In this section, we limit the incorporation of the effect of sediments to the deformation model by making use of a sediment amplification curve, extracted from elastodynamic simulations of layered sediment-rock seabed [6].

9. When the bathymetries from several sources are merged, special care needs to be paid because they may have different datum. Please explain how the author handled this issue.

Reply – We thank the referee for highlighting this important methodological aspect. We have clarified our approach and current limitations in Appendix B.

10. How do the authors define ‘high-definition’ of resolution in bathy, mesh, and coastline? It seems it refers $O(10m)$, but there have been similar achievements up to those resolutions so far and it is not a quite challenging issue.

Reply – We are grateful to the referee for this comment, since it has helped to sharpen the main thrust of the work. We have clarified this aspect, and our main challenges in the second and last paragraphs of Section 1. Although ‘high-definition’ primarily means the $O(10m)$ mesh resolution in the vicinity of the port, it also touches on the resolutions of the bathymetries which are comparable to that of the mesh. Indeed, we agree with the referee that there exist tsunami current simulations at $O(10m)$ (and even for $O(5m)$). Despite possible issues arising from handling fine resolutions, our main challenge lies in encapsulating a large number of these high-definition simulations within a statistical framework. This is an essential requirement for probabilistic hazard assessments and stretches the limit of current High-performance Computing (HPC) facilities, even with the latest GPU (Graphics Processing Unit) acceleration. Hence, we argue that our framework of statistical emulation in this work, pushes the boundaries of current state-of-the-art in quantifying port hazard – with multi-threaded emulation platform for large-scale (1 million) predictions, built on 300 high-definition simulations on smart unstructured meshes (10m), using massively parallel multi-GPU-enabled simulations of latest tsunami models, and hierarchical file formats – all integrated in an overarching workflow.

11. Figure 2: It is not easy to catch the main flow of Figure 2. Further simplification and conceptualization are required for the elegant representation. Also, the section number included in the figure didn’t match with one

in the main body.

Reply – We are extremely thankful to the referee for pointing out the typographical errors in the flowchart. We have incorporated the changes as suggested, which have greatly simplified the flowchart (now Figure 1a). Further, the detailed and corrected version of the flowchart is included as supplementary material.

13. Please cite Yuan et al. (2020) where applicable, since it provided good accomplishment of using GPU in tsunami modeling.

Reply – The above reference has been cited in the first paragraph of Section 2(c).

Jaffe B, Goto K, Sugawara D, Gelfenbaum G, La Selle S. 2016. Uncertainty in tsunami sediment transport modeling. *Journal of Disaster Research* 11(4): 647–661.

Li L, Qiu Q, Huang Z. 2012. Numerical modeling of the morphological change in Lhok Nga, west Banda Aceh, during the 2004 Indian Ocean tsunami: understanding tsunami deposits using a forward modeling method. *Natural Hazards* 64: 1549–1574.

Lynett, P. J., Gately, K., Wilson, R., Montoya, L., Arcas, D., Aytore, B., ... & David, C. G. (2017). Inter-model analysis of tsunami-induced coastal currents. *Ocean Modelling*, 114, 14-32.

Yuan, Y., Shi, F., Kirby, J. T., & Yu, F. (2020). FUNWAVE-GPU: Multiple-GPU Acceleration of a Boussinesq-Type Wave Model. *Journal of Advances in Modeling Earth Systems*, 12(5), e2019MS001957.

Zhang, X., & Niu, X. (2020). Probabilistic tsunami hazard assessment and its application to southeast coast of Hainan Island from Manila Trench. *Coastal Engineering*, 155, 103596.

References

- [1] Lynett PJ et al.. 2017 Inter-model analysis of tsunami-induced coastal currents. *Ocean Modelling* **114**, 14 – 32.

- [2] Giles D, Kashdan E, Salmanidou DM, Guillas S, Dias F. 2020 Performance analysis of Volna-OP2 – massively parallel code for tsunami modelling. *Computers & Fluids* **209**, 104649.
- [3] Li L, Qiu Q, Huang Z. 2012 Numerical modeling of the morphological change in Lhok Nga, west Banda Aceh, during the 2004 Indian Ocean tsunami: understanding tsunami deposits using a forward modeling method. *Natural Hazards* **64**, 1549–1574.
- [4] Jaffe B, Goto K, Sugawara D, Gelfenbaum G, Selle SL. 2016 Uncertainty in Tsunami Sediment Transport Modeling. *Journal of Disaster Research* **11**, 647–661.
- [5] Okada Y. 1985 Surface deformation due to shear and tensile faults in a half-space. *Bulletin of the Seismological Society of America* **75**, 1135–1154.
- [6] Dutykh D, Dias F. 2010 Influence of sedimentary layering on tsunami generation. *Computer Methods in Applied Mechanics and Engineering* **199**, 1268 – 1275. Multiscale Models and Mathematical Aspects in Solid and Fluid Mechanics.

Appendix B

RESPONSE TO REFEREE #2

The manuscript presents a method to evaluate the probabilistic hazard of tsunami currents in specific target sites (e.g., harbours), using statistical emulators to reduce the computational resources. Moreover, the work is focused on tsunami currents which present some differences in terms of hazard with respect the more common use of tsunami wave heights as hazard intensity measure.

The topic is of interest for the tsunami community and I agree with authors that both the method (statistical emulator) and the kind of intensity measure (tsunami currents) are still not much explored in tsunami science.

However, I have some comments that, in my opinion, should be addressed before the publications.

Reply – The authors thank the referee for the review, *esp.* for the comment on the need for the intensity measure and statistical method as explored in this work. Addressing the referee’s comments have helped in enriching the content of the manuscript, whilst also enhancing its structure.

Please find a point-by-point reply below. Major modifications to passages in the revised manuscript are shown in red. An exception is when whole sections/sub-sections have undergone considerable modification, in which case the section/sub-section title appears in red.

First of all, I have a very general comment.

My main concern is the possible misleading of the actual goal of the manuscript: in its present form, the work is illustrative of a general workflow that can be used to assess probabilistic tsunami hazard. Each step is technically detailed in its implementation and this is a really worthy effort from the authors. But, on the other hand, I would stress that they should be careful to avoid that a reader could be confused that this could be an example of hazard assessment. To be clearer: immediately in the abstract it is stated that the work ” [...] map probabilistic representations of maximum tsunami velocities and heights for the full range of possible future tsunamis at around 200 locations around Karachi port, Makran Subduction Zone (MSZ)”. In my opinion, this is not true. A rigorous source analysis from the seismic and tectonic point of view is missing, so I disagree with authors that the ”full range” of possibilities is here considered. For this reason, I suggest that in some part of the text (likely in the title itself? ”A method to quantify... ” or something like this?) authors should try to stress more that this work is

purely illustrative of a method.

My thoughts are partially confirmed by the authors themselves: indeed, in the discussion, they are more prudent, with a list of missing sources of uncertainty that should be considered before performing "a full probabilistic assessment" (page 21, lines 51 and following): scaling relations, G-R maximum moment magnitudes, bathymetry. So they are in part aware of this.

Reply – We are thankful that the referee finds the exposition of the implementation worth appreciating. We agree with the ambiguous connotations of the mentioned phrase "full range" in the abstract. Thus, we have removed the phrase not only from the abstract, but from the rest of the text as well. In the third paragraph of the introduction, we have cast the statistical emulation framework as another probabilistic route among many. Additionally, the nature/limitation of the proposed statistical emulation framework as a PTHA methodology is extensively described in Section 4, starting with the new sentence – "Although, a full/complete probabilistic description of hazard may remain elusive, a realistic goal of 'fullness' will be to carefully define and perform each step in the PTHA. In these terms, a 'full' probabilistic assessment would ideally need to include further sources of uncertainties, including a thorough analysis of the source uncertainties in its seismic and tectonic setting. ...".

Moreover, I think that also the seismic source variability is only partially explored and poorly described; in my opinion, this is an essential component of such hazard analyses and should at least be presented. Indeed, even though the case study is illustrative, the MSZ source zone and the Karachi Port are real, so I think that it is important to introduce the geophysical context and better address some input choices.

I'm not asking to explicitly explore the full source variability: but I think that authors should, at least, add an exhaustive section to introduce the tectonic setting of the area and better explain that they are considering only a part of the potential sources for illustrative purposes.

Reply – Taking into consideration the referee's comment, Section 2(a) now starts with a description of the tectonic setting of the MSZ.

Here, for example, the authors are excluding the western part of the MSZ subduction, which is already a quite important choice (see my comment #1). Then, what about other potential tsunamigenic seismic sources? Could splay faults, outer-rise earthquakes and crustal earthquakes generate tsunamis with impact for the Karachi port? Could these sources be easily incorporated in the present procedure or they represent a challenge? Please enrich the text with this part.

Reply –

Choice of eastern MSZ - We thank the referee for pointing out this aspect, and agree that an illustration of the entire MSZ would have been more appealing. We have modified the text in the first paragraph of Section 2(a) to reflect this limitation. Given the scope of this work, we limit ourselves to the eastern MSZ. This is mainly because tsunamis from western MSZ would have less appreciable effects on Karachi port than those arising from the western MSZ. Further, paleoseismic accounts hypothesize that the western MSZ is seismically inactive compared to the eastern MSZ.

Splay, outer-rise and crustal earthquakes - We acknowledge that these non-standard earthquake scenarios can be included with additional parameterisation, a description has been included in Section 4. Although a large increase in the number of parameters (especially for spatial fields of parameters) presents a challenge to emulation, a solution presents itself in the combination of dimension reduction and emulation.

An important future step (mentioned in Section 4) would be to model the entire MSZ for an area-wide assessment of hazard due to velocities, *i.e.* at the major ports in Pakistan, Iran, Oman, and India, whilst accounting for crustal, outer-rise and imbricate faults.

Finally, some subjective choices are just presented as they are, without explanations: this could lead a reader to think that these could be common choices, but they aren't (see my comment #2 about the range of magnitudes).

Reply – We thank the referee for pointing out these lacunae. The choices are explained in the response to comment #2 below.

Other comments:

1. Why excluding the western part of MSZ? It seems to me that the illustrative example could be more solid if applied to the whole subduction instead to a part of it. The considered G-R distribution refers to the whole subduction or just to the eastern part? Could you please address this? I wasn't able to find Table S1 (mentioned at page 17, lines 34-35) so I could not check how the authors selected the b-value. However, if this extension to the western MSZ does not present technical issues, why do not model it? The results would be representative of the whole subduction and could be more interesting to comment.

Reply – The reason for including only the eastern MSZ has been explained in a reply above.

Choice of b-value - We thank the referee for pointing out the discrepancy

in the citation, it has been rectified. Further, the discussion below is now reflected in Section 3(c). The b -value of 0.92 is sourced from Table S2 in the electronic supplementary material for the Earthquake Model of Middle East (EMME) database [1]. Although this refers to the whole MSZ, we assume it as representative for the eastern part. This is reasonable given the scope of our work. Further, any changes in the parameters of the G-R relation (*i.e.* β , M_w^m , M_w^M *etc.*) only affect the earthquake samples generated for the prediction stage of the emulation. These changes can be handled in a very efficient manner as the prediction stage is the cheapest component in the entire workflow. In fact, cheap prediction permits fast propagation of uncertainties in the G-R parameters to the hazard intensities. Here, we demonstrate this for two values of one such parameter, the maximum magnitude M_w^M ($= 8.8$ and 8.6). All of the above description has been incorporated into Section 3(c).

2. Why the minimum considered magnitude is 7.5? for saving numbers because of illustrative purposes or there is any geophysical reason to neglect magnitudes < 7.5 ? Small tsunamis couldn't generate currents as well? Please address this point.

Reply – We thank the referee for this important question. We have addressed this concern in Sections 3(b) and 3(c). The minimum magnitude of 7.5 has been considered for illustrative purposes. Further, the L-O-O (Leave-One-Out) diagnostics show that some of the lower M_w events do not generate appreciable velocities. In these cases, their location with respect to the port is such that negligible wave energy is radiated to the port.

3. I found that the state of the art is not complete. For example, other works describing general methods to propagate the uncertainty and/or to reduce the number of tsunami simulations required to perform probabilistic tsunami hazards have been published (Lorito et al., 2015, Volpe et al., 2019 and very recently Kotani et al., 2020 could be of interest). I think that, even though it was on landslide-generated tsunami, the work on statistical emulators by Sarri et al. in 2012 represents an existing reference of a similar approach and merit be mentioned.

Reply – We thank the referee for suggesting these important references. They have been incorporated in the third paragraph of Section 1 where probabilistic methods for tsunami hazard assessment are reviewed.

4. Authors could re-think the manuscript's structure, for example by moving to Appendix some of the most technical parts, in order to lighten the reading of the main text, which is now very dense of figures and sub-figures that could distract the reader. One explicit example is the following: Appen-

dices A, B and C have a corresponding section with the same title also in the main text (Slip profile generation, Merging of Bathymetries and Localised Non-Uniform Unstructured Mesh); I think that this is redundant: I suggest to keep the Appendices only and also move there the corresponding figures.

Reply – The referee’s suggestion has helped in streamlining the manuscript. The main text has been lightened by integrating the former technical/less innovative sections related to slip profile generation, merging of bathymetries and mesh generation into Appendices A, B, and C respectively. The corresponding figures have also been relocated. Further, in the spirit of the referee’s suggestion, all the figures have been extensively reworked to refine the layout, decrease clutter and increase visibility.

5. This is just a suggestion for a future implementation: a further module that could be incorporated in the procedure is the effect of the water layer between the seafloor displacement and the corresponding water surface perturbation (see Kajiura, 1963).

Reply – We thank the referee for this suggestion. We have included this aspect in Section 4.

Some minor comments:

- I can’t find any table mentioned in the text. Maybe there was an issue during files’ submission?

Reply – We clarify that there is no table in the manuscript. But there are two tables referred to in the text, and both occur in the references cited. To further clarify, both the citations have been edited.

- Introduction, page 2, line 24: A $M_w=7.7$ earthquake is not small. I think authors can remove the adjective ”small” which is not necessary (and not understandable) in this sentence.

Reply – Section 1 has been modified, so this is no longer required.

- Introduction, page 2, line 26: I cannot find Table 1 and reference to Figure 3 seems not correct, please check.

Reply – Section 1 has been modified, so this is no longer required.

- Introduction, page 2, line 28: I suggest to use labels when referring to subplots (ad example Figure 1b) to help the reader.

Reply – Figure 1 and Section 1 have been modified, so this is no longer required. Care has been taken to precisely refer to the other figures/subfigures in the manuscript.

- Why, in figures 4, 5 and 9, the 3000 m depth contour line is explicitly shown? It seems to me a little confusing with the coastline. Is it necessary in these plots? If not, you could remove it and keep only the colormap in figure 7a for water depth.

Reply – We comply with the referee’s suggestion. The 3000 m depth contour has been removed from all figures.

Note on supplementary material – Two supplementary animations have been created to show the exact finite-fault configuration, slip profile, and seabed deformation for the 300 cases (with and without the influence of sediments). They also portray how the Latin hypercube sampling sweeps through the input parameter space, and the manner in which the dimensions of the 300 sources are scaled with respect to the scaling relation. Also, the flowchart in the main text has been simplified, and an extended flowchart added as supplementary material.

References:

Kajiura K., 1963. The leading wave of a tsunami, *Bull. Earthq. Res. Inst. Univ. Tokyo*, 41, 535–571

Kotani T., K. Tozato, S. Takase, S. Moriguchi, K. Terada, Y. Fukutani, Y. Otake, K. Nojima, M. Sakuraba, Y. Choe, Probabilistic tsunami hazard assessment with simulation-based response surfaces, *Coastal Engineering*, 160, 2020, doi:10.1016/j.coastaleng.2020.103719

Lorito S., Selva, J., Basili, R., Romano, F., Tiberti, M. M., & Piatanesi, A. (2015). Probabilistic hazard for seismically induced tsunamis: Accuracy and feasibility of inundation maps. *Geophysical Journal International*, 200(1), 574–588. <https://doi.org/10.1093/gji/ggu408>

Sarri, A., Guillas, S., and Dias, F.: Statistical emulation of a tsunami model for sensitivity analysis and uncertainty quantification, *Nat. Hazards Earth Syst. Sci.*, 12, 2003–2018, <https://doi.org/10.5194/nhess-12-2003-2012>, 2012

Volpe, M., Lorito, S., Selva, J., Tonini, R., Romano, F., and Brizuela, B.: From regional to local SPTHA: efficient computation of probabilistic tsunami inundation maps addressing near-field sources, *Nat. Hazards Earth Syst. Sci.*, 19, 455–469, <https://doi.org/10.5194/nhess-19-455-2019>, 2019.

References

- [1] Danciu L, Sesetyan K, Demircioglu M et al.. 2018 The 2014 Earthquake Model of the Middle East: seismogenic sources. *Bulletin of Earthquake Engineering* **16**, 3465–3496.